# SSD: A Sparse Semantic Defense Against Semantic Adversarial Attacks to Image Classifiers

## Abstract

Adversarial attacks to image classifiers pose a major threat to machine learning models. However, existing defenses against such attacks have been designed mostly for unrealistic image threat models, such as bounded $\ell_p$-norm image perturbations. In this paper, we focus on defending against more realistic *semantic adversarial attacks*, which modify semantic image concepts (e.g., make it in snow) that are irrelevant to the underlying classification task (e.g., classify a dog). Intuitively, a classifier that is robust to semantic attacks should rely only on concepts that are relevant for the task. Therefore, the proposed Sparse Semantic Defense (SSD) uses large language models to build a dictionary of visual concepts that are relevant for a given visual recognition task, and large vision-language models to embed images and concepts into an aligned, shared latent space. Sparse coding is then used to decompose the image embedding as a sparse combination of the text embeddings of relevant concepts plus a residual term that captures irrelevant concepts, including semantic attacks. We provide a theoretical justification for why sparse coding can separate irrelevant semantics from the resulting sparse code. A simple linear classifier on the sparse code is then used. SSD is also interpretable by design because it relies on task-relevant visual concepts. Experiments on ImageNet show that SSD performs favorably with respect to other baselines in terms of robust accuracy against semantic adversarial attacks while maintaining interpretability.

## 1 Introduction

Deep neural network classifiers are vulnerable to adversarial attacks, i.e., imperceptible perturbations to their input that can alter the classifier's prediction (Szegedy et al., 2013). Existing methods for computing such adversarial attacks, such as the Fast Gradient Sign Method (FGSM) (Goodfellow et al., 2014), Projected Gradient Descent (PGD) (Madry et al., 2019), and Carlini-Wagner (C&W) attacks (Carlini and Wagner, 2017), assume that such perturbations are bounded in $\ell_p$-norms. The success of such attacks motivated the development of various defense mechanisms (Cohen et al., 2019; Madry et al., 2019; Nie et al., 2022; Shafahi et al., 2019; Wong et al., 2020). Among them, the most popular are adversarial training (Madry et al., 2019; Shafahi et al., 2019; Wong et al., 2020), randomized smoothing (Cohen et al., 2019; Pautov et al., 2022), and input purification (Nie et al., 2022). These defenses are effective against $\ell_p$ attacks with a trade-off being the high cost of additional computation, either at training (with adversarial training) or at inference time (with randomized smoothing or input purification).

However, the assumption of $\ell_p$-bounded attacks overlooks critical vulnerabilities to larger, semantically coherent, and content-preserving perturbations (Gilmer et al., 2018), which better reflect sophisticated real-world manipulations. While early semantic attacks like geometric transformations (Hsiung et al., 2023) or hue-shifting (Hosseini and Poovendran, 2018) were a step in this direction, they could produce unnatural images. Recent works leverage powerful generative models to craft highly natural and challenging *semantically meaningful* perturbations (Hsiung et al., 2023; Joshi et al., 2019; Liu et al., 2023; Qiu et al., 2020; Shamsabadi et al., 2020; Wang et al., 2023). These methods modify human-interpretable attributes (e.g., background scenery) that preserve the image's content, but can effectively deceive classifiers. Among these, Instruct2Attack (Liu et al., 2023) is most applicable to diverse image classification tasks. It employs a text-conditioned diffusion model to implement specific, classification-irrelevant semantic changes (such as weather or time of day), optimizing these natural-looking modifications to fool a given classifier effectively.

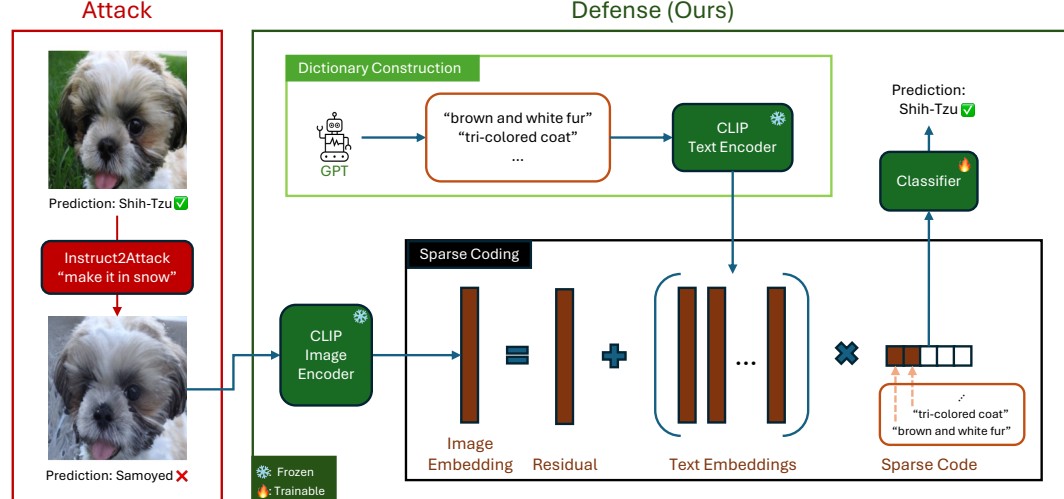

Figure 1: **Our approach to defending against semantic attacks.** After constructing a set of classification-relevant semantic concepts, we calculate their CLIP text embeddings to get a dictionary of concepts. Next, we use efficient sparse coding algorithms to decompose the CLIP embedding of a test image as a sparse linear combination of these concepts plus a residual term that captures irrelevant concepts and semantic attacks. Finally, we use a linear classifier on this sparse code to get the final prediction. This provides interpretability and robustness against semantic adversarial attacks that mainly affect classification-irrelevant semantic concepts while preserving classification-relevant ones.

Defending against these stronger types of semantic attacks introduces *two* challenges. **First**, traditional defenses (such as adversarial training, randomized smoothing, and input purification) face significant efficiency issues. Adversarial training becomes computationally infeasible due to the high cost of generating semantic adversarial examples for training (e.g., from models like Instruct2Attack that require backpropagation through diffusion processes) and the combinatorial explosion of possible semantic directions. Other approaches, such as randomized smoothing and input purification, typically incur substantial inference time overhead. **Second**, existing defenses do not contribute to a better understanding of adversarial vulnerability. Given that semantic attacks are inherently interpretable, an interpretable defense mechanism could provide crucial insights into which features are relevant for classification versus those that are spurious and exploited by attacks. Current defenses predominantly improve the robustness of black-box models without offering such interpretability, thereby hindering a deeper understanding of model failure modes and the development of more targeted defenses. These aforementioned challenges motivate the following research question:

> How can we develop an *efficient* and *interpretable* defense mechanism for semantic attacks?

To answer this question, we propose Sparse Semantic Defense (SSD), a sparse representation-based classifier defined on a natural language semantic concept space. SSD leverages pre-trained vision-language models like Contrastive Language-Image Pre-training (CLIP) Radford et al. (2021), which provide a joint vision-language embedding space in which an image embedding can be represented as a sparse combination of text concept embeddings. Building on this, we represent an image as a sparse linear combination of classification-relevant concepts plus a residual term that subsumes irrelevant concepts, including those induced by semantic attacks. This offers natural robustness because semantic attacks typically preserve classification-relevant concepts in the semantic space while altering irrelevant ones. We summarize our algorithm and highlight the contributions below, with an overview in Figure 1:

1. First, **we develop a novel method to construct a semantic concept dictionary** from CLIP text embeddings, leveraging GPT (Achiam et al., 2023) and the WordNet hierarchy (Miller, 1995) to achieve a balance between expressivity and interpretability in representing visual attributes (Section 3.2). In particular, given the set of WordNet visual objects (which includes the ImageNet classes), we use GPT to generate the concepts describing these objects.

2. Second, we propose a simple classifier that is both robust to semantic attacks and interpretable. Because the image of an object can be described using a small number of concepts in the dictionary, **we use efficient sparse coding algorithms to represent an image embedding as a sparse linear combination of concept embeddings** (Section 3.3). The sparse coefficients is then fed to a linear classifier (Section 3.4), allowing direct identification of salient concepts influencing predictions.

3. Third, we provide theoretical justification for our approach, showing that **the sparse coding step effectively filters out irrelevant concepts, including those altered by semantic attacks**, thereby preserving classification-relevant information (Proposition 1).

4. Fourth, **we empirically demonstrate through experiments on ImageNet that our approach achieves competitive robust accuracy** against semantic adversarial attacks, while simultaneously offering significantly enhanced interpretability of model decisions compared to existing defenses.

## 2 PRELIMINARIES

### 2.1 ADVERSARIAL ATTACKS AND DEFENSE

An $\ell_p$ adversarial attack can be formulated as finding an $\ell_p$-norm bounded input perturbation that changes the prediction of a classifier $f$ (Goodfellow et al., 2014; Madry et al., 2019)

$$\min_{\boldsymbol{x}'} \|\boldsymbol{x}' - \boldsymbol{x}\|_p \leq \nu \quad \text{subject to} \quad f(\boldsymbol{x}') \neq f(\boldsymbol{x}), \tag{1}$$

where $\boldsymbol{x}$ and $\boldsymbol{x}'$ are the original and corrupted images, respectively, with an upper bound of $\nu$ on the difference in some $\ell_p$-norm ($l_2$ and $l_\infty$ are the most common). Since $\ell_p$ attacks are well-studied, there have been many defense strategies (Cohen et al., 2019; Madry et al., 2019; Nie et al., 2022; Shafahi et al., 2019; Wong et al., 2020). Among these defenses, adversarial training, which augments the training samples with $\ell_p$-attacked samples, has been the most popular due to its effectiveness. However, adversarial training requires fresh adversarial examples at each training step, making it a costly solution (Shafahi et al., 2019; Wong et al., 2020).

A semantic adversarial attack generalizes $\ell_p$ attacks by measuring perturbations in a semantic space:

$$\min_{\boldsymbol{x}'} d(\boldsymbol{x}', \boldsymbol{x}) \leq \nu \quad \text{subject to} \quad f(\boldsymbol{x}') \neq f(\boldsymbol{x}), \tag{2}$$

where $d(\cdot, \cdot)$ is a distance metric in a semantic space (e.g., the LPIPS distance (Zhang et al., 2018)). Initial works focus on specific semantic aspects, such as hue/saturation (Hosseini and Poovendran, 2018), brightness (Hsiung et al., 2023), and color/texture (Bhattad et al., 2019; Shamsabadi et al., 2020). A recent work develops a more general framework to generate semantic attacks through a text-conditioned diffusion model (Liu et al., 2023). Unlike $\ell_p$ attacks, there has not been much interest in developing a defense against semantic attacks. A simple attempt would be adversarial training with semantic adversarial examples. However, this approach is computationally infeasible due to the high cost of generating semantic adversarial examples (e.g., by backpropagating through a diffusion process (Liu et al., 2023)) and the combinatorial number of semantic directions.

We focus on semantic attacks for *three* reasons. **First**, $\ell_p$ attacks have been well-studied in the literature and there are specified defenses for $\ell_p$ attacks (interested readers are referred to Costa et al. (2024) and Long et al. (2022)). **Second**, $\ell_p$ attacks are not realizable in the real world, while semantic attacks are. Thus, defending against semantic attacks is more practical. **Third**, $\ell_p$ attacks and semantic attacks are simply two different kinds of attacks and the solution to one might not transfer to the other. As a consequence, they should be treated differently until we can find a universal solution to both.

### 2.2 REPRESENTATIVE SEMANTIC ADVERSARIAL ATTACKS

The earliest approach to designing semantic attacks was modifying the hue and saturation of the image (Hosseini and Poovendran, 2018; Hsiung et al., 2023). Later works, such as ColorFool (Shamsabadi et al., 2020), focus on modifying the color/texture of the region of the images in a semantically meaningful range. A more general approach is Instruct2Attack, which employs a general-purpose, text-conditioned diffusion model (Brooks et al., 2022), enabling broader applicability. Given each clean image $\boldsymbol{x}$, a pretrained text-conditioned image-editing diffusion model (Brooks et al., 2022),

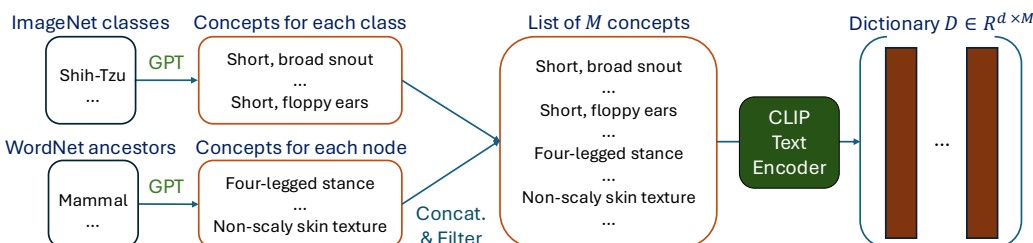

Figure 2: Dictionary construction with GPT given a list of class names and the ancestors (more general synsets) of the classes in the WordNet hierarchy (Miller, 1995), where $d$ is the embedding dimension of CLIP (Radford et al., 2021).

and a fixed text prompt $L$, Instruct2Attack parameterizes some guidance vectors in the diffusion process of this diffusion model to edit the clean image into an adversarial image that maximizes the classification loss of a victim classifier $f$ while keeping the image perceptually similar to the clean image (See Appendix B for details).

## 3 METHOD

In this section, we derive an efficient and interpretable defense to semantic adversarial attacks. To achieve this goal, we leverage the power of vision-language models to decompose an image into a sparse linear combination of classification-relevant text concepts in a semantic concept dictionary, plus a residual term for irrelevant concepts and semantic attacks. This allows us to build an interpretable classifier that depends only on the decomposition coefficients. In Section 3.1, we briefly review how dictionaries of semantic concepts are created by previous work. In Section 3.2, we show how to construct this dictionary of semantic concepts given the class names for a classification task. In Section 3.3, we describe how to construct the sparse codes that define the semantic concepts present in the image, which can be efficiently computed in practice. In Section 3.4, we present how to construct an interpretable classifier given the sparse codes. The full algorithm is shown in Algorithm 1.

### 3.1 PRIOR ARTS ON SEMANTIC CONCEPT DICTIONARY CONSTRUCTION

Constructing a dictionary of semantic concepts has a long history in machine learning and computer vision. Early methods rely on human-annotated concepts for each dataset (Koh et al., 2020; Kumar et al., 2009; Lampert et al., 2009), but this is not very scalable, especially as the number of classes is large (e.g., ImageNet (Deng et al., 2009) has 1000 classes). Oikarinen et al. (2023) propose a method that uses a large language model (LLM) to generate concepts for each class, which is more scalable. Yang et al. (2023) propose to also prune the concepts using submodular optimization to maximize coverage of the concept space. Chiquier et al. (2024) combine evolutionary algorithms with an LLM to generate the concepts. We take the same basic approach of using an LLM to help generate the concepts, but take it a step further by using the hierarchy information in WordNet (Miller, 1995) to construct a set of coarse-to-fine concepts.

### 3.2 HIERARCHICAL CONSTRUCTION OF SEMANTIC DICTIONARY

Towards our goal of expressing images as linear combinations of various text concepts, it is critical to first create a high-quality dictionary of semantic text concepts. We would like this dictionary to be both expressive and interpretable. Expressivity requires covering many distinct class-relevant features so that different images can be represented accurately using the vast set of concepts. Interpretability requires maintaining visual and monosemantic concepts in the dictionary so that the underlying text representations of an image are interpretable to an end user.

To this end, for a given classification task with $K$ class names, we hierarchically generate concepts following a coarse-to-fine strategy. Specifically, we utilize the hierarchical graph of the nodes in WordNet, where the leaf nodes are the classes names [1]. We traverse the nodes in this graph and instruct

---

[1]A visualization of this hierarchy is publicly available at https://observablehq.com/@mbostock/imagenet-hierarchy

---

**Algorithm 1** Classification via Semantic Decomposition

---

**Require:** Class names $\{c_k\}_{k=1}^K$, CLIP image encoder $\mathcal{E}_I$, CLIP text encoder $\mathcal{E}_T$, training data
$\{(\boldsymbol{x}_i, y_i)\}_{i=1}^N$, WordNet hierarchy $G$ ($\{c_k\}_{k=1}^K \subseteq G$)
    *// Step 1: Dictionary Construction (Section 3.2)*
1: $\mathcal{C} \leftarrow \emptyset$                                                   ▷ Initialize concept set
2: **for** $g = 1$ to $G$ **do**
3:     $\mathcal{C}_g \leftarrow$ Generate concepts for node $g$ using GPT-4
4:     $\mathcal{C} \leftarrow \mathcal{C} \cup \mathcal{C}_g$
5: **end for**
6: $\mathcal{C} \leftarrow$ Filter concepts (Appendix D.4)
7: $\mathbf{D} \leftarrow [\mathcal{E}_T(c) \text{ for } c \in \mathcal{C}]$                                ▷ Dictionary $\in \mathbb{R}^{d \times |\mathcal{C}|}$
    *// Step 2: Sparse Feature Construction (Section 3.3)*
8: **for** $i = 1$ to $N$ **do**
9:     $\boldsymbol{z}_i \leftarrow \arg\min_{\boldsymbol{z}} \|\mathcal{E}_I(\boldsymbol{x}_i) - \mathbf{D}\boldsymbol{z}\|_2^2 + \lambda\|\boldsymbol{z}\|_1$              ▷ Solve LASSO
10: **end for**
    *// Step 3: Linear Classifier Construction (Section 3.4)*
11: $\mathbf{W} \leftarrow \arg\min_{\mathbf{W}} \sum_{i=1}^N \mathcal{L}_{ce}(\mathbf{W}^T\boldsymbol{z}_i, y_i)$
12: **return** Classifier $f(\boldsymbol{x}) = \arg\max_k [\mathbf{W}^T\boldsymbol{z}]_k$

---

GPT-4 (Achiam et al., 2023) to generate concepts for each node while maintaining discriminability with sibling nodes (see our exact prompt in Figure 5).

Concatenating these concepts gives us a large set of expressive semantic concepts. To ensure the diversity of the concepts, we also apply a concept filtering step, similar to Oikarinen et al. (2023), which can be interpreted as increasing the incoherence of the dictionary, a property that is useful for sparse coding Foucart et al. (2013). The specific details of this filtering step are given in Appendix D.4.

Finally, given this list of text concepts, we use the CLIP text encoder to compute the corresponding text embeddings. This gives us our final dictionary of semantic concepts $\mathbf{D} \in \mathbb{R}^{d \times M}$, where $d$ is the dimension of the CLIP embedding space, and $M$ is the number of text concepts.

### 3.3 CONSTRUCTING SPARSE FEATURES

Having constructed the dictionary of concepts $\mathbf{D}$, we now formulate the problem of recovering relevant concepts as a sparse coding problem (Foucart et al., 2013). The overview of our method is shown in Figure 1. Given a CLIP image embedding $\boldsymbol{x}$, we use sparse coding to decompose it as a sparse combination of the $M$ dictionary concepts in $\mathbf{D}$, computed by solving the LASSO optimization problem:

$$\min_{\boldsymbol{z}} \|\boldsymbol{x} - \mathbf{D}\boldsymbol{z}\|_2^2 + \lambda\|\boldsymbol{z}\|_1, \tag{3}$$

where $\lambda$ is the sparsity parameter. We use the LASSO formulation since it encourages sparsity, thus promoting further interpretability by selecting only a few highly relevant semantic concepts per image. The result of this step is a sparse code $\boldsymbol{z} \in \mathbb{R}^M$ for each image embedding $\boldsymbol{x}$.

If the dictionary constructed in the previous subsection is expressive enough, this sparse code should capture all relevant concepts for a particular image, while the residual term of this sparse coding problem captures irrelevant concepts (such as those modified by semantic attacks). We formalize this intuition in the following proposition.

**Proposition 1.** *Let $\mathbf{D} \in \mathbb{R}^{d \times M}$ ($d < M$) have unit-norm columns and satisfy the Restricted Isometry Property (RIP) of order $4S$ with constant $\delta_{4S} < 1/3$ (Candes et al., 2006). Suppose $\boldsymbol{z}_0 \in \mathbb{R}^M$ is $S$-sparse, and we observe*

$$\boldsymbol{x} = \mathbf{D}\boldsymbol{z}_0 + \boldsymbol{e}, \tag{4}$$

*where the noise $\boldsymbol{e} \in \mathbb{R}^d$ is orthogonal to the column space of $\mathbf{D}$, i.e. $\mathbf{D}^\top \boldsymbol{e} = 0$. Then the solution $\boldsymbol{z}^\star$ of Basis Pursuit Denoising*

$$\min_{\boldsymbol{z}} \|\boldsymbol{z}\|_1 \quad s.t. \quad \|\boldsymbol{x} - \mathbf{D}\boldsymbol{z}\|_2 \leq \|\boldsymbol{e}\|_2$$

*satisfies $\boldsymbol{z}^\star = \boldsymbol{z}_0$ and $\boldsymbol{x} - \mathbf{D}\boldsymbol{z}^\star = \boldsymbol{e}$.*

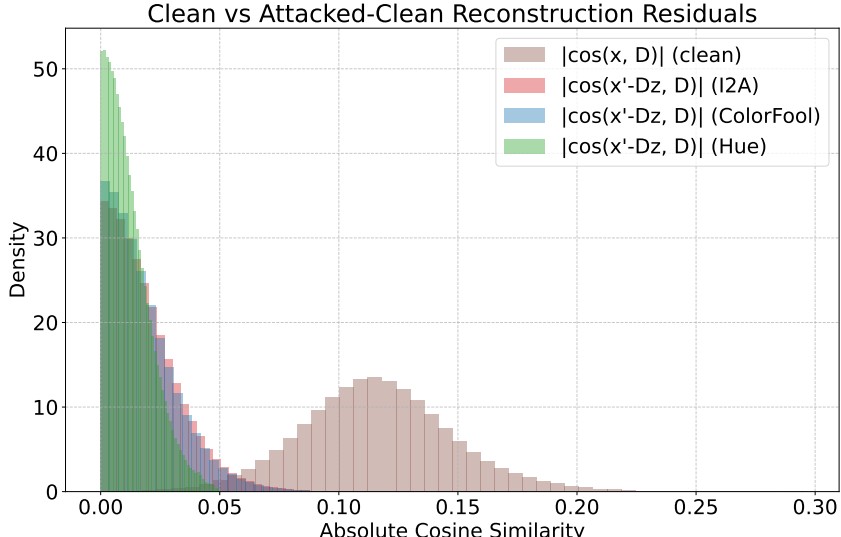

Figure 3: **The residual between the attacked embedding and the clean reconstruction is not correlated with the dictionary atoms.** The analysis is done on the dataset ImageWoof with Instruct2Attack (Liu et al., 2023), ColorFool (Shamsabadi et al., 2020), and Hue & Saturation (Hsiung et al., 2023). This supports our claim that the dictionary captures all relevant concepts, and the attack only modifies irrelevant concepts.

The proof is provided in Appendix A. Formally, if the noise generated by the semantic attack is orthogonal to the column space of $\mathbf{D}$ (i.e., $\cos(\boldsymbol{e}, \mathbf{D}) = \cos(\boldsymbol{x}' - \mathbf{D}\boldsymbol{z}, \mathbf{D}) \approx 0$)[2], the dictionary satisfies the RIP condition, and the original sparse code is sufficiently sparse, then the sparse code computed by solving equation 3 should be close to the clean sparse code (i.e., $\boldsymbol{z}' \approx \boldsymbol{z}$). In other words, if the dictionary is expressive enough and the attack only perturbs the irrelevant concepts, then the attacked sparse code should be close to the original sparse code. We verify that this is true for semantic attacks in Figure 3. This shows that our method can effectively filter out semantic adversarial attacks.

### 3.4 Constructing an Interpretable Classifier

Finally, given the sparse codes $\{\boldsymbol{z}_i\}_{i=1}^N$, we train a linear classifier $\mathbf{W} \in \mathbb{R}^{M \times K}$ to predict the final output. The classifier is trained by solving the following optimization problem:

$$\min_{\mathbf{W}} \sum_{i=1}^N \mathcal{L}_{ce}(\mathbf{W}^T \boldsymbol{z}_i, y_i),$$ (5)

where $\mathcal{L}_{ce}$ is the cross-entropy loss, and $y_i$ is the ground truth label of the $i$-th image. The learned weight matrix $\mathbf{W}$ intuitively represents how strongly each semantic concept contributes to each class prediction, providing interpretability by design.

## 4 Experiments

We evaluate the robustness against semantic adversarial attacks and the interpretability of our method for image classification on ImageNet. We first show that our method performs competitively, demonstrating that our approach effectively enhances robustness in Section 4.1. We then analyze the stability of the sparse codes and the interpretability of the model's decision in Section 4.2. The full experimental details can be found in Appendix C.

---

[2]We note that a result relaxing this condition to quantify only approximate orthogonality (through incoherence) instead of exact orthogonality exists in Cai et al. (2009).

Table 1: **ImageWoof Results: SSD increases robust accuracy against adaptive semantic adversarial attacks without any adversarial training.** Comparison of standard and robust accuracy for different models on ImageWoof dataset across different semantic attacks. The baselines are CLIP (Radford et al., 2021), FARE2-CLIP (Mao et al., 2022), and TeCoA2-CLIP (Schlarmann et al., 2024). The attacks are Instruct2Attack (Liu et al., 2023) and ColorFool (Shamsabadi et al., 2020).

| Model | Std. Acc. ↑ | ColorFool Rob. Acc. ↑ | I2A Rob. Acc. ↑ |
|---|---|---|---|
| CLIP | **92.5** | 53.7 | 20.0 |
| FARE2-CLIP | 81.4 | 54.3 | 44.7 |
| TeCoA2-CLIP | 84.5 | 46.9 | 49.0 |
| CLIP + SSD (Ours) | 87.6 | **62.0** | **53.5** |

Table 2: **ImageNet Results: SSD increases robust accuracy against adaptive semantic adversarial attacks without any adversarial training.** The baselines are CLIP (Radford et al., 2021) and FARE2-CLIP (Schlarmann et al., 2024). The attacks are Instruct2Attack (Liu et al., 2023), Hue & Saturation (Hsiung et al., 2023), and ColorFool (Shamsabadi et al., 2020).

| Model | Std. Acc. ↑ | Hue & Sat. Rob. Acc. ↑ | ColorFool Rob. Acc. ↑ | I2A Rob. Acc. ↑ |
|---|---|---|---|---|
| CLIP | **82.7** | 73.1 | 15.7 | 6.0 |
| FARE2-CLIP | 80.9 | 72.5 | 19.9 | 8.9 |
| CLIP + SSD (Ours) | 81.7 | **77.1** | **27.3** | **12.8** |

**Datasets.** We evaluate our method on two datasets: ImageWoof (Howard, 2019) and ImageNet (Russakovsky et al., 2015). ImageWoof is a subset of ImageNet with 10 classes of dogs, making it smaller for quick evaluation while still having classes similar enough to each other to be challenging. All images are resized to $256 \times 256 \times 3$. We use the full training sets for all the datasets. On ImageNet, for the attack evaluation, we use a subset of 5000 test images used by RobustBench (Croce et al., 2021).

**Attacks.** As stated in Section 2.1, we focus on semantic attacks and thus evaluate the results against semantic attacks for the experiments. A more detailed description of the attacks considered are provided in Section 2.2. The first attack we consider is Instruct2Attack (Liu et al., 2023), a state-of-the-art semantic attack using diffusion models to edit the clean image adversarially according to a text prompt. We use the same attack settings as in the original paper and the fixed prompts "make it in snow" and "make it at night" for all images. Additionally, we also evaluate our method against two other semantic attacks: ColorFool (Shamsabadi et al., 2020) and Hue & Saturation (Hsiung et al., 2023). We use the official implementations of these attacks with default settings.

**Method.** To solve the sparse coding problem in Equation 3, we use the LASSO solver in the spams package (Mairal et al., 2014). We set the maximum number of non-zero elements to be 100 on ImageWoof, and 200 on ImageNet, unless otherwise specified.

### 4.1 ROBUSTNESS AGAINST SEMANTIC ADVERSARIAL ATTACKS

**Baselines.** We first compare our method with the most popular defense method, adversarial training. In particular, we consider adversarial training on $\ell_2$ (Mao et al., 2022; Schlarmann et al., 2024). Then, we also consider a popular diffusion-based input purification method called DiffPure (Nie et al., 2022). The backbone is ViT-L/14 (Dosovitskiy et al., 2020) for all models. Attacks against our methods and all baselines (except for DiffPure (Nie et al., 2022)) are adaptive attacks. Due to the high computational cost of generating adaptive attacks for DiffPure (Nie et al., 2022), we evaluate DiffPure with non-adaptive attacks, which are attacks calculated against a CLIP model with a linear classifier.

**Results.** We report the standard accuracy and robust accuracy against adaptive attacks on ImageWoof and ImageNet in Tables 1 and 2, respectively. Our method consistently improves robust accuracy over the baselines using $\ell_2$ adversarial training. Additionally, we compare the results on a non-adaptive attacks to DiffPure (Nie et al., 2022) in Table 4. Although DiffPure (Nie et al., 2022) has a slight improvement in accuracy compared to SSD, SSD is much faster.

**Comparison with other dictionaries.** We compare our dictionary with other dictionaries in Table 11. We see that our method achieves the best balance between standard and robust accuracy on ImageWoof.

## 4.2 INTERPRETABILITY AND STABILITY OF THE SPARSE CODE

First, we qualitatively evaluate interpretability by visualizing the most influential semantic concepts for a correctly/incorrectly classified image from ImageWoof with the prompt "make it in snow" in Figure 7 and 8, respectively. We can see that the concepts are visual, semantic, and relevant to the object in the picture. An interesting observation is that when the top concepts change minimally from the clean to the attacked image, the prediction is more likely to be the same. To quantitatively evaluate this observation, we plot the difference between the standard and robust accuracy conditioned on the intersection over union (IoU) between the clean and attacked sparse code on ImageWoof with the prompt "make it in snow" in Figure 4. As IoU increases, the difference between standard and robust accuracy decreases linearly, showing that one reason adversarial attacks succeed is by perturbing the semantic meaning of the input image. Importantly, we only gain this insight from the interpretable classifier that we constructed.

## 5 RELATED WORK

**Adversarial attacks.** Adversarial attacks start with $\ell_p$-norm bounded attacks, which are implemented using with FGSM (Goodfellow et al., 2014), PGD (Madry et al., 2019), C&W (Carlini and Wagner, 2017), and AutoAttack (Croce and Hein, 2020). Because assuming that the perturbation is bounded in $\ell_p$ norms is unrealistic, there is a movement toward more larger but more semantically meaningful attacks. Early attempts modify either hue/saturation (Hosseini and Poovendran, 2018), brightness (Hsiung et al., 2023), color/texture (Bhattad et al., 2019). However, these attacks are not semantically meaningful and are unnatural to humans. In contrast, semantic adversarial attacks (Joshi et al., 2019; Liu et al., 2023; Qiu et al., 2020; Wang et al., 2023) only modify semantic components of the image, and the new image look natural. However, most methods specifically focus on face recognition tasks (Joshi et al., 2019; Qiu et al., 2020; Wang et al., 2023), limiting their generality. In contrast, Instruct2Attack (Liu et al., 2023) employs a general-purpose text-conditioned diffusion model applicable to diverse image classification tasks.

**Adversarial defenses.** The most popular defense mechanism against adversarial attacks is adversarial training (Kurakin et al., 2016; Laidlaw et al., 2020; Madry et al., 2019). Adversarial training augments the training set with adversarial examples from one or more attacks. As a result, adversarial training has a computational cost proportional to the cost of generating the adversarial examples, making it challenging to scale to expensive attacks. Another category includes methods like randomized smoothing (Cohen et al., 2019; Pautov et al., 2022), which provide certified robustness by analyzing the consensus of predictions on noisy input variations. Finally, input purification techniques (Nie et al., 2022) remove adversarial perturbations prior to classification, often leveraging auxiliary generative models. However, both randomized smoothing and input purification techniques requires a high degree of extra computation at test time. In comparison, our method requires a smaller overhead at test time.

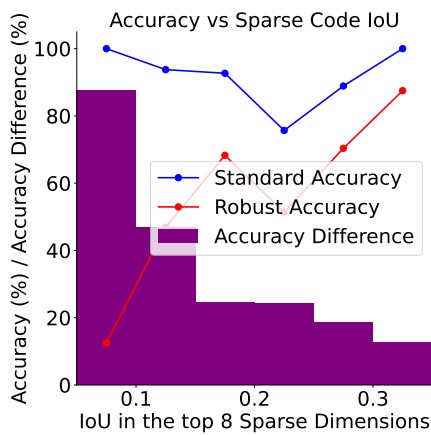

Figure 4: **The model's predictions are more stable as the sparse codes are stable**. We plot the difference between standard/robust accuracy conditioned on the IoU between clean-attacked sparse codes.

**Sparse representation.** Although sparse representations were explored extensively in early deep learning research (Coates and Ng, 2011; Kavukcuoglu et al., 2010; Zeiler et al., 2010), dense representations have since become dominant (He et al., 2016; Krizhevsky et al., 2012; Vaswani et al., 2017). However, there is a resurgence of interest in integrating sparse representations to leverage their potential for controllability,

efficiency, and even semantic decomposition (Chattopadhyay et al., 2023b; Luo et al., 2024; Wen et al., 2025). Sparse coding has also been explored in the context of adversarial robustness (Thaker et al., 2022) and explainability (Chattopadhyay et al., 2023b). While Thaker et al. (2022) only considers sparse coding in the input space to counter $\ell_p$ attacks, Chattopadhyay et al. (2023b) considers the connection between sparse coding and information pursuit for explainability. However, these approaches differ significantly from our focus on semantic attacks.

**Semantic decomposition.** There has been a growing interest in interpretable by design methods by using semantic concepts (Kim et al., 2018; Koh et al., 2020). This line of work starts with Concept Bottleneck Models (Chen et al., 2020; Koh et al., 2020; Kumar et al., 2009; Lampert et al., 2009), in which the dictionary of concepts is manually constructed. Subsequent works (Chiquier et al., 2024; Oikarinen et al., 2023; Yang et al., 2023) show that the dictionary can be automatically constructed with LLMs. Another line of work (Chattopadhyay et al., 2023a;b) approaches semantic decomposition from the perspective of information pursuit. To our knowledge, our approach is the first to counter semantic adversarial attacks by leveraging semantic decomposition.

# 6 DISCUSSION

## 6.1 LIMITATIONS

A limitation of our work is that the fact that we need a vision-language model (e.g., CLIP) as the backbone. In most vision tasks, this is not a big problem since vision-language models are readily available (Cherti et al., 2023). However, for a specific domain (e.g., medical images), one would need a more specialized vision-language model. Another limitation is that the concept generated by GPT-4 might not align well with the semantic concepts that CLIP can detect from images, which can lead to suboptimal performance. Finally, since we rely on relevant concepts for classification, if an adversary modifies the relevant concepts of the object adversarially (e.g., elongating the facial shape of a cat to make it look like a dog), then our method is not guaranteed to perform well. However, such an attack is not content-preserving (Gilmer et al., 2018) and is out of the scope of this work.

## 6.2 FUTURE WORK

Our work opens several interesting questions and directions for future research in adversarial robustness and interpretability. To generate the concepts for the dictionary, we need a taxonomy of the relations between concepts, such as WordNet (Miller, 1995). Note that the full ImageNet (Deng et al., 2009) only labels 21,841 out of 80,000 synsets in WordNet, so class names outside of ImageNet might still be in WordNet. Application of our method to other domains might require a different taxonomy (e.g., RadLex (Langlotz, 2006) for radiology concepts). While semantic attacks like Instruct2Attack (Liu et al., 2023) have proven highly effective at generating semantically meaningful adversarial examples, their significant computational demands limit practical applications, including adversarial training. Thus, an important open question is creating more efficient semantic adversarial attacks.

# 7 CONCLUSION

In this paper, we present an efficient and interpretable defense mechanism for semantic adversarial attacks. Our method constructs a dictionary of semantic concepts and leverages efficient sparse coding algorithms to represent images as sparse combinations of these concepts. By classifying the resulting sparse codes with a linear classifier, our model naturally provides interpretability, enabling quick identification of the critical concepts influencing each prediction. Through experiments on ImageNet, we show that our method significantly improves the robust accuracy of CLIP-based image classification models. Through extensive experiments on ImageNet, we demonstrate that our method significantly enhances the robust accuracy of CLIP-based image classification models while preserving interpretability. Our approach thus bridges robustness and interpretability, offering a promising direction for defending against semantic adversarial threats. By making machine learning systems, particularly image classifiers, more resilient to semantic attacks, this work contributes to enhancing the trustworthiness and reliability of machine learning applications.

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

CONTENTS

## A    PROOF OF PROPOSITION 1

*Proof.* For any $z$, writing $h = z - z_0$ gives

$$x - Dz = e - Dh. \tag{6}$$

Since $D^\top e = 0$, we have $e \perp \mathrm{col}(D)$, and therefore

$$\|x - Dz\|_2^2 = \|e\|_2^2 + \|Dh\|_2^2 \geq \|e\|_2^2, \tag{7}$$

with equality iff $Dh = 0$. Thus $z_0$ yields the minimal feasible residual norm. The fact that $z_0$ is the unique minimizer follows from the nullspace property implied by RIP. Specifically, if RIP holds, then no nonzero vector supported on at most $2S$ indices lies in the nullspace of $D$. Hence no nontrivial $h$ with $\|z_0 + h\|_1 \leq \|z_0\|_1$ can satisfy $Dh = 0$, and the $\ell_1$ minimizer is unique. Therefore $z^\star = z_0$ and the residual equals $e$. $\qquad\square$

## B    DETAILS ON THE GUIDANCE VECTORS IN INSTRUCT2ATTACK

Latent diffusion models (Rombach et al., 2021) take the original image $x$ and a pure noise vector $z_T$, progressively denoise it following a noise schedule parameterized by $\sigma_t$, output the denoised latent vector $z_0$, and use a decoder to map $z_0$ to a desired image $x'$. The denoising process progressive denoises $z_T$ as follows:

$$z_{t-1} = z_t + (\sigma_t^2 - \sigma_{t-1}^2)\tilde{\epsilon}_\theta(z_t, x, c_L, \alpha, \beta) + \sqrt{\frac{\sigma_{t-1}^2(\sigma_t^2 - \sigma_{t-1}^2)}{\sigma_t^2}}\zeta_t, \tag{8}$$

where $\xi_t \sim \mathcal{N}(0, I)$ and $\tilde{\epsilon}_\theta$ is the score function of the diffusion model. This score is conditioned on the original image $x$, the image-editing text prompt $c_L$, and two guidance vectors $\alpha$ and $\beta$. Finally, an image decoder maps the final latent vector $z_0$ to the adversarial image $x'$. Finding an adversarial example is equivalent to optimizing $\alpha$ and $\beta$ given each clean image $x$:

$$\max_{\alpha,\beta} \mathcal{L}_{ce}(f(g_{\alpha,\beta}(x)), y) - \lambda \max\left(0, d(g_{\alpha,\beta}(x), x) - \gamma\right), \tag{9}$$

where $\mathcal{L}_{ce}$ is the cross-entropy loss, and $d(\cdot, \cdot)$ denotes the LPIPS distance (Zhang et al., 2018), an automated metric designed to quantify perceptual similarity between images as judged by humans, and $\gamma$ is the perturbation budget. Intuitively, the result from the optimization problem is a modified image that is most likely to change the prediction of a classifier (by maximizing the cross entropy loss) while keeping the perturbed image perceptually similar to the original image (by minimizing the LPIPS distance).

## C    EXPERIMENT SETTINGS

In this section, we provide additional details on the experiment settings. In Section C.1, we provide details on the resources used for the experiments. In Section C.2, we provide details on the hyperparameters used for the attacks.

### C.1    RESOURCES

We conduct experiments on a server with 8 NVIDIA A5000 GPUs. We use PYTORCH and fix a seed whenever possible to ensure reproducibility. The statistics of the datasets used in this paper are shown in Table 3.

### C.2    ATTACK SETTINGS

The hyperparameters we use for Instruct2Attack, in their notation (Liu et al., 2023, Section 4.1), are $\lambda = 100$, $\gamma = 0.3$, $\eta = 0.1$, $T = 20$, $s_f = 1.5$, $s_r = 7.5$, and $N = 200$. If the InstructPix2Pix (Brooks et al., 2022) base model is too expensive to run on more modest resources, we believe that using UIP2P (Simsar et al., 2024) as a more efficient alternative is a good option, although neither Instruct2Attack (Liu et al., 2023) nor we tested this alternative.

For Hue & Saturation attacks (Hsiung et al., 2023), we use the range of $[-\pi, \pi]$ for the hue and $[0.7, 1.3]$ for the saturation. We run separate attacks for hue and saturation and take the average in the result.

Table 3: Dataset Sizes and Number of Classes for Training and Test Sets.

| Dataset | Training Set Size | Test Set Size | Number of Classes |
|---------|-------------------|---------------|-------------------|
| ImageWoof | 8,687 | 162 | 10 |
| ImageNet | 1,281,167 | 50,000 | 1,000 |

## C.3 MODELS

We use CLIP ViT-L/14 as the base model for all experiments in this paper.

To train the linear classifier, we use AdamW for $600$ epochs with a learning rate of $10^{-4}$ on both ImageWoof and ImageNet. We use a batch size of $1024$.

# D SSD DETAILS

## D.1 BACKPROPAGATION THROUGH THE SPARSE CODE

To generate adaptive attacks against our method, we need to compute the gradient of the input through the sparse coding step. Following Mairal et al. (2011), the gradient with respect to the input is given by:

$$\nabla_{\mathbf{x}} f(\mathbf{D}, \mathbf{x}) = \mathbf{D}\boldsymbol{\beta}^*, \tag{10}$$

where $\boldsymbol{\beta}^*$ is computed as follows:

$$\boldsymbol{\beta}^*_{\Lambda^C} = 0 \quad \text{and} \quad \boldsymbol{\beta}^*_{\Lambda} = (\mathbf{D}_\Lambda^\top \mathbf{D}_\Lambda)^{-1} \nabla_{\boldsymbol{\alpha}_\Lambda} \ell_{ce}(\mathbf{y}, \mathbf{W}, \boldsymbol{\alpha}^*), \tag{11}$$

and $\boldsymbol{\alpha}^*$ is the sparse code solution from Equation 3, $\Lambda$ contains the indices of non-zero elements in $\boldsymbol{\alpha}^*$, $\mathbf{D}$ is the dictionary matrix, $\mathbf{D}_\Lambda$ is the submatrix of $\mathbf{D}$ containing only columns corresponding to non-zero elements in $\boldsymbol{\alpha}^*$, $\ell_{ce}$ is the cross-entropy loss function, $\mathbf{y}$ is the one-hot encoded target label, and $\mathbf{W}$ is the weight matrix of the linear classifier.

## D.2 FULL RESULTS

We provide the full results for all the models for the "make it in snow" attack prompts in Table 5 (adaptive attacks) and Table 6 (non-adaptive attacks), and for the "make it at night" attack prompts in Table 7 (adaptive attacks) and Table 8 (non-adaptive attacks). We find that our method is only behind DiffPure (Nie et al., 2022) in terms of robust accuracy while being more efficient. One thing to note is that our method is the first defense designed specifically for semantic attacks.

We provide more details on the FARE and TeCoA defenses in the following. FARE2 and FARE4 means that a CLIP model was adversarially trained with the FARE method (Schlarmann et al., 2024) under $\ell_\infty$ attacks with $\epsilon = 2/255$ and $\epsilon = 4/255$, respectively. [3] Similarly, TeCoA2 and TeCoA4 means that a CLIP model was adversarially trained with the TeCoA method (Mao et al., 2022) under $\ell_\infty$ attacks with $\epsilon = 2/255$ and $\epsilon = 4/255$, respectively. Note that the base CLIP model in these experiments is still the ViT-L/14 model to ensure consistency. It is interesting that the models trained with both FARE/TeCoA has a degree of robustness to the attacks, but a larger adversarial training radius hurts the robust accuracy against semantic attacks. As such, we only include the results for these defenses with the adversarial training radius of $\epsilon = 2/255$ in Table 1 and Table 2.

We also provide the details of how we use DiffPure (Nie et al., 2022). [4] We use the VP-SDE model with default hyperparameters provided in their code [5].

---

[3]Pretrained models can be found at `https://github.com/chs20/RobustVLM`

[4]Code can be found at `https://github.com/NVlabs/DiffPure`

[5]`https://github.com/NVlabs/DiffPure/blob/master/configs/imagenet.yml`

---

**GPT-4 Prompt for Dictionary Construction**

TASK: Generate at least {num_concepts} distinctive visual characteristics for the category
"{class_name}" that can:
1. Describe most instances of this category
2. Differentiate it from other categories, especially: {similar_classes}
3. Focus on visual appearance only (not behavior, habitat, or non-visual properties)
4. Be used in a fine-grained classification task such as ImageNet
HIERARCHY CONTEXT: This category belongs to the following hierarchy path:
{hierarchy_path}
This means that it shares some visual characteristics with the categories along this path while
having some unique characteristics.
ANCESTOR CONCEPTS: The following are concepts already generated for ancestor cate-
gories. Generate more specific and distinctive concepts for this category and avoid repeating
these general concepts:
{ancestor_concepts}
INCLUDE CHARACTERISTICS FROM THESE CATEGORIES:
- Shape and structure (overall form, proportions)
- Surface features (texture, patterns)
- Color variations and distinctive markings
- Key parts and their appearance (e.g., eyes, legs, tail)
- Distinctive poses or typical orientations
- Characteristic details that aid identification from similar categories
AVOID:
- Background elements like "snow", "grass", or "night sky"
- Non-visual properties like weight, behavior, or habitat
- Overly generic descriptors that could apply to many categories
- Hedging language like "often", "sometimes", "typically"
- Background/Environment: (e.g., Grass, Snow, Sand, City skyline, Wooden floor, Brick wall)
- Lighting/Atmosphere: (e.g., Nighttime, Sunset, Cloudy sky, Fog, Spotlight, Overexposure)
- Camera/Photographic Effects: (e.g., Low-angle shot, Bokeh, Motion blur, Black-and-white
filter, Lens flare)
- Temporary states or context: (e.g., Rain droplets, Wind-blown leaves, Puddle)
GOOD CHARACTERISTICS:
- "Square muzzle" (specific shape)
- "White-tipped tail" (specific marking)
- "Tricolor coat" (distinctive color pattern)
- "Dense double coat" (specific texture)
POOR CHARACTERISTICS:
- "Dog-like appearance" (too vague)
- "Often found in homes" (not visual)

Figure 5: **Prompt used for constructing the dictionary of image characteristics.** {num_concepts}
is the number of concepts to generate (which we set to 30 in our experiments), {class_name} is the
name of the class/synset, {similar_classes} is a list of classes that are similar to the class, taken as
the 3-generation cousins of the class, {hierarchy_path} is the hierarchy path from the node to the
root of the WordNet hierarchy, {ancestor_concepts} is list of concepts already generated for ancestor
categories (and to be avoided).

## D.3 ADDITIONAL EXPERIMENTAL RESULTS

**Ablation Study on Sparsity Level.** We investigate the impact of different sparsity levels on both
standard and robust accuracy on ImageWoof. Figure 6 shows the results for various maximum
numbers of non-zero elements in the sparse code. We observe that increasing the sparsity level
generally improves both standard and robust accuracy, with diminishing returns beyond 100 elements.

Table 4: **Comparison of standard and robust accuracy for non-adaptive attacks on ImageWoof and ImageNet.** Time denotes the average inference time.

| Dataset | Model | Time (s) ↓ | Standard Accuracy ↑ | Robust Accuracy ↑ |
|---|---|---|---|---|
| ImageWoof | CLIP (Radford et al., 2021) | **0.01** | **92.5** | 20.0 |
| | CLIP + DiffPure (Nie et al., 2022) | 11.3 | 85.1 | **67.2** |
| | CLIP + SSD (Ours) | 0.36 | 87.6 | 60.1 |
| ImageNet | CLIP (Radford et al., 2021) | **0.01** | **82.7** | 6.0 |
| | CLIP + DiffPure (Nie et al., 2022) | 11.3 | 74.6 | **34.1** |
| | CLIP + SSD (Ours) | 0.81 | 81.7 | 29.9 |

Table 5: **Comparison of standard and robust accuracy for different models on ImageWoof and ImageNet with adaptive attacks with "make it in snow" prompt.**

| Dataset | Model | Standard Accuracy ↑ | Robust Accuracy ↑ |
|---|---|---|---|
| ImageWoof | CLIP (Radford et al., 2021) | **92.5** | 25.9 |
| | FARE2-CLIP (Schlarmann et al., 2024) | 81.4 | 42.5 |
| | TeCoA2-CLIP (Mao et al., 2022) | 84.5 | 48.7 |
| | CLIP + SSD (Ours) | 85.8 | **56.1** |
| ImageNet | CLIP (Radford et al., 2021) | **82.7** | 1.1 |
| | FARE2-CLIP (Schlarmann et al., 2024) | 80.9 | 4.9 |
| | FARE4-CLIP (Schlarmann et al., 2024) | 77.5 | 4.3 |
| | TeCoA2-CLIP (Mao et al., 2022) | 80.5 | **5.2** |
| | TeCoA4-CLIP (Mao et al., 2022) | 76.2 | 3.8 |
| | CLIP + SSD (Ours) | 81.7 | 5.1 |

**Ablation Study on the Optimizer for the Linear Classifier.** We compare the standard and robust accuracy of our method with different optimizers for the linear classifier on ImageWoof. We consider AdamW (Loshchilov and Hutter, 2019), ISTA+SAGA (Wong et al., 2021), and FISTA (Beck and Teboulle, 2009). The results are shown in Table 9. We see that using AdamW to optimize the linear classifier yields the best robust and clean accuracy.

## D.4 ADDITIONAL DETAILS ON DICTIONARY CONSTRUCTION

We use `gpt-4o-2024-11-20` to generate the dictionary for each synset in WordNet (Miller, 1995) (including the ImageNet classes). The prompt is shown in Figure 5. We note that reason we use GPT-4 is that previous work (Newell and Rosenbloom, 1981; Oikarinen et al., 2023) have used GPT models and we don't have any reason to believe that other models would perform better. If the cost of GPT's API is an issue, open-source models like Llama 3 (AI@Meta, 2024) and DeepSeek-v3 (Liu et al., 2024) are also good candidates, although we have not tested them.

We show examples of the concepts generated for each synset in Table 13. As we go from more general (e.g., Animal) to more specific (e.g., Shih-Tzu), the concepts become more fine-grained and specific to the category. Our full dictionary is included in the supplementary material.

After generating the dictionary using GPT-4, we perform a post-processing step to filter out similar concepts, which reduces the number of concepts from 35,280 to 15,905. Note that this step is first introduced in Oikarinen et al. (2023). The number of concepts remaining after each step is shown in Table 12. The steps we take are as follows:

1. **Filter concepts that are too similar to class names.** The main reason for this step is to prevent the model from explaining the image using class names, which is trivial and non-informative. We use the CLIP text encoder to compute the similarity between the concept and the class name. If the similarity is above 0.9, we filter out the concept. Formally, given a set of concepts $\mathcal{C}$ and class names $\{c_k\}_{k=1}^{K}$, we filter out concepts that are too similar

Table 6: **Comparison of standard and robust accuracy for different models on ImageWoof and ImageNet with "make it in snow" attack (Non-Adaptive Attacks).**

| Dataset | Model | Standard Accuracy ↑ | Robust Accuracy ↑ |
|---|---|---|---|
| ImageWoof | CLIP (Radford et al., 2021) | **92.5** | 25.9 |
| | DiffPure (Nie et al., 2022) + CLIP | 85.1 | **74.0** |
| | CLIP + SSD (Ours) | 85.8 | 67.9 |
| ImageNet | CLIP (Radford et al., 2021) | **82.7** | 1.1 |
| | DiffPure (Nie et al., 2022) + CLIP | 74.6 | **38.1** |
| | CLIP + SSD (Ours) | 81.7 | 35.7 |

Table 7: **Comparison of standard and robust accuracy for different models on ImageWoof and ImageNet with adaptive attacks with "make it at night" prompt.**

| Dataset | Model | Standard Accuracy ↑ | Robust Accuracy ↑ |
|---|---|---|---|
| ImageWoof | CLIP (Radford et al., 2021) | **92.5** | 14.2 |
| | FARE2-CLIP (Schlarmann et al., 2024) | 81.4 | 46.9 |
| | TeCoA2-CLIP (Mao et al., 2022) | 84.5 | 49.3 |
| | CLIP + SSD (Ours) | 85.8 | **50.9** |
| ImageNet | CLIP (Radford et al., 2021) | **82.7** | 10.9 |
| | FARE2-CLIP (Schlarmann et al., 2024) | 80.9 | 13.0 |
| | FARE4-CLIP (Schlarmann et al., 2024) | 77.5 | 12.4 |
| | TeCoA2-CLIP (Mao et al., 2022) | 80.5 | 13.5 |
| | TeCoA4-CLIP (Mao et al., 2022) | 76.2 | 11.9 |
| | CLIP + SSD (Ours) | 81.7 | **20.5** |

to class names with the following formulation:

$$\mathcal{C}_1 = \{c \in \mathcal{C} \mid \max_{k=1,\ldots,K} \cos(\mathcal{E}_T(c), \mathcal{E}_T(c_k)) < 0.9\} \tag{12}$$

where $\cos(\cdot, \cdot)$ denotes the cosine similarity between two vectors.

2. **Filter out concepts that are too similar to each other.** We also use the CLIP text encoder to compute the similarity between all pairs of concepts. As stated before, this step can be interpreted as increasing the incoherence of the dictionary (Foucart et al., 2013). If the similarity is above 0.9, we filter out the concept with a higher cosine similarity to the other concept, keeping the more informative concept. Formally, we filter out concepts that are too similar to each other with the following formulation:

$$\mathcal{C}_2 = \{c \in \mathcal{C}_1 \mid \max_{c' \in \mathcal{C}_1 \setminus \{c\}} \cos(\mathcal{E}_T(c), \mathcal{E}_T(c')) < 0.9\} \tag{13}$$

The final filtered set of concepts is $\mathcal{C}_2$, which is used to construct the dictionary matrix $\mathbf{D}$.

### D.5 ADDITIONAL VISUALIZATIONS

We first provide a visualization of the top concepts when the prediction is correct (Figure 7) and when the prediction is incorrect (Figure 8). We see that the top concepts are more stable when the prediction is correct, i.e., more concepts are shared between the clean image and the adversarial image. This suggests that our method is able to capture meaningful concepts that are relevant to the classification task. Additionally, we also provide visualizations of the sparse codes in Figure 10 (for Oikarinen et al. (2023)), Figure 11 (for Gandelsman et al. (2024)), and Figure 12 (for Chiquier et al. (2024)). Our dictionary is more focused on the visual attributes of the images, which is more interpretable than the other dictionaries.

Table 8: **Comparison of standard and robust accuracy for different models on ImageWoof and ImageNet with "make it at night" attack (Non-Adaptive Attacks).**

| Dataset | Model | Standard Accuracy ↑ | Robust Accuracy ↑ |
|---|---|---|---|
| ImageWoof | CLIP (Radford et al., 2021) | **92.5** | 14.2 |
| | DiffPure (Nie et al., 2022) + CLIP | 85.1 | **60.4** |
| | CLIP + SSD (Ours) | 85.8 | 52.2 |
| ImageNet | CLIP (Radford et al., 2021) | **82.7** | 10.9 |
| | DiffPure (Nie et al., 2022) + CLIP | 74.6 | **30.2** |
| | CLIP + SSD (Ours) | 81.7 | 24.1 |

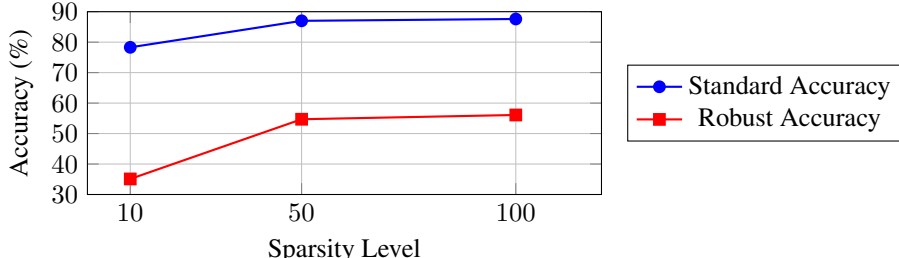

Figure 6: Comparison of standard and robust accuracy for different sparsity levels (maximum number of non-zero elements) on ImageWoof. Higher sparsity levels generally improve both standard and robust accuracy, with diminishing returns beyond 100 elements.

# E  UPDATED RESULTS

## E.1  EMPIRICAL RESULTS

We provide additional visualizations of the orthogonality between the noise and the dictionary in Figure 13 for ImageNet.

To test the generality of our method to different prompts, we use the remaining standard prompts provided by Liu et al. (2023) to provide additional results on ImageWoof in Table 14 and Table 15.

Table 9: **Comparison of standard and robust accuracy for different weight constraint mechanisms on ImageWoof.** Results show that AdamW provides the best robust accuracy while different L1 optimization methods yield varying performance.

| Optimizer | Standard Acc. ↑ | Robust Acc. ↑ |
|---|---|---|
| AdamW (Loshchilov and Hutter, 2019) | **85.8** | **56.1** |
| ISTA+SAGA (Wong et al., 2021) | 80.2 | 14.1 |
| FISTA (Beck and Teboulle, 2009) | 87.0 | 3.7 |

Table 10: **Our dictionary with hierarchy achieves the best balance between standard and robust accuracy.** The dictionary we use in the main text is the Visual concepts + Hierarchy dictionary based on this result. We find that this dictionary achieves the best balance between standard and robust accuracy.

| Dictionary | Standard Acc. ↑ | Robust Acc. ↑ |
|---|---|---|
| Oikarinen et al. (2023) | **88.2** | 41.9 |
| Chiquier et al. (2024) | 61.7 | 17.2 |
| Gandelsman et al. (2024) | 81.4 | 20.3 |
| Non-visual concepts (Ours) | 86.4 | 56.1 |
| Visual concepts (Ours) | 83.9 | **58.6** |
| Visual concepts + Hierarchy (Ours) | 87.6 | 56.1 |

Table 11: **Our dictionary achieves the best balance between standard and robust accuracy.**

| Dictionary | Std. Acc. ↑ | I2A Rob. Acc. ↑ |
|---|---|---|
| Oikarinen et al. (2023) | **88.2** | 41.9 |
| Chiquier et al. (2024) | 61.7 | 17.2 |
| Gandelsman et al. (2024) | 81.4 | 20.3 |
| Ours | 87.6 | **56.1** |

Table 12: Number of concepts, incoherence, and coverage of the dictionary at different steps.

| Step | Number of Concepts | Incoherence ↓ | Coverage ↑ |
|---|---|---|---|
| After generating the dictionary with GPT-4 | 35,280 | 1.000 | 1.000 |
| After filtering out concepts that are too similar to class names | 25,773 | 0.996 | 0.999 |
| After filtering out concepts that are too similar to each other | 15,905 | 0.899 | 0.965 |

Table 13: Example concepts for our dictionary across synsets in WordNet and ImageNet classes.

| Synset | Example Concepts |
|--------|------------------|
| **Animal** (A synset in WordNet) | "Vibrant feather colors"
"Segmented exoskeleton"
"Flattened fins"
"Striped fur pattern"
"Rounded shell"
"Prominent tusks"
"Spotted coat markings"
"Horns or antlers" |
| **Mammal** (A synset in WordNet) | "Distinctive fur patterns"
"Non-scaly skin texture"
"Color variation in coat"
"Visible mammary glands"
"Four-legged stance"
"Short or elongated fur"
"Rounded paws or hooves" |
| **Canine** (A synset in WordNet) | "Tricolor fur pattern on body"
"Defined forehead stop on face"
"Prominent facial mask markings"
"Color gradient on fur coat" |
| **Shih-Tzu** (an ImageNet class) | "Dark eye rims enhancing expression"
"Fur parted along the spine"
"Bushy, curled tail carried high"
"Short, broad snout with wrinkles"
"Long fur covering the legs"
"White and gold fur combination"
"Flat, wide face with short muzzle"
"Square-shaped head proportions"
"Short, floppy, feathered ears" |

Table 14: **Comparison of standard and robust accuracy for different models on ImageWoof and ImageNet with adaptive attacks with the "make it a sketch painting" prompt.**

| Dataset | Model | Standard Accuracy ↑ | Robust Accuracy ↑ |
|---------|-------|---------------------|-------------------|
| ImageWoof | CLIP (Radford et al., 2021) | **92.5** | 32.7 |
| | FARE2-CLIP (Schlarmann et al., 2024) | 81.4 | 38.8 |
| | CLIP + SSD (Ours) | 85.8 | **51.2** |
| ImageNet | CLIP (Radford et al., 2021) | **82.7** | 4.44 |
| | FARE2-CLIP (Schlarmann et al., 2024) | 80.9 | 4.47 |
| | CLIP + SSD (Ours) | 81.7 | **9.40** |

Table 15: **Comparison of standard and robust accuracy for different models on ImageWoof with adaptive attacks with the "make it a vintage photo" prompt.**

| Dataset | Model | Standard Accuracy ↑ | Robust Accuracy ↑ |
|---------|-------|---------------------|-------------------|
| ImageWoof | CLIP (Radford et al., 2021) | **92.5** | 27.7 |
| | FARE2-CLIP (Schlarmann et al., 2024) | 81.4 | 44.4 |
| | CLIP + SSD (Ours) | 85.8 | **53.7** |

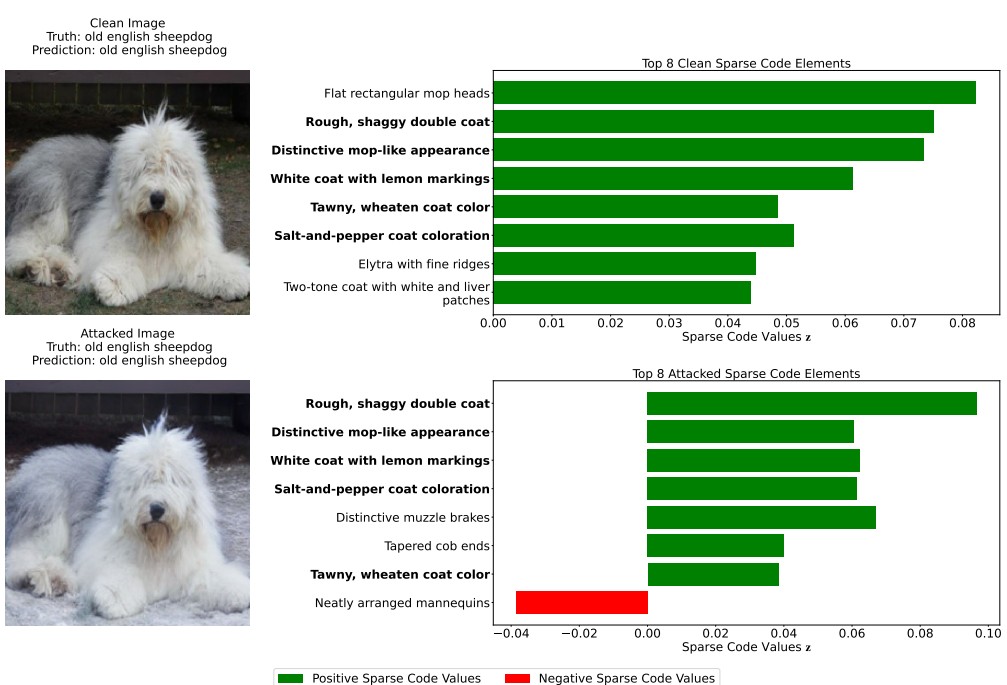

Figure 7: **The top concepts are stable when the prediction is correct**. We visualize the top 8 concepts given by our method, sorted based on the absolute value of the product of the sparse codes with the true label classifier weights. Concepts that are present in both the sparse code corresponding to the clean image and the sparse code corresponding to the adversarial image are in **bold**.

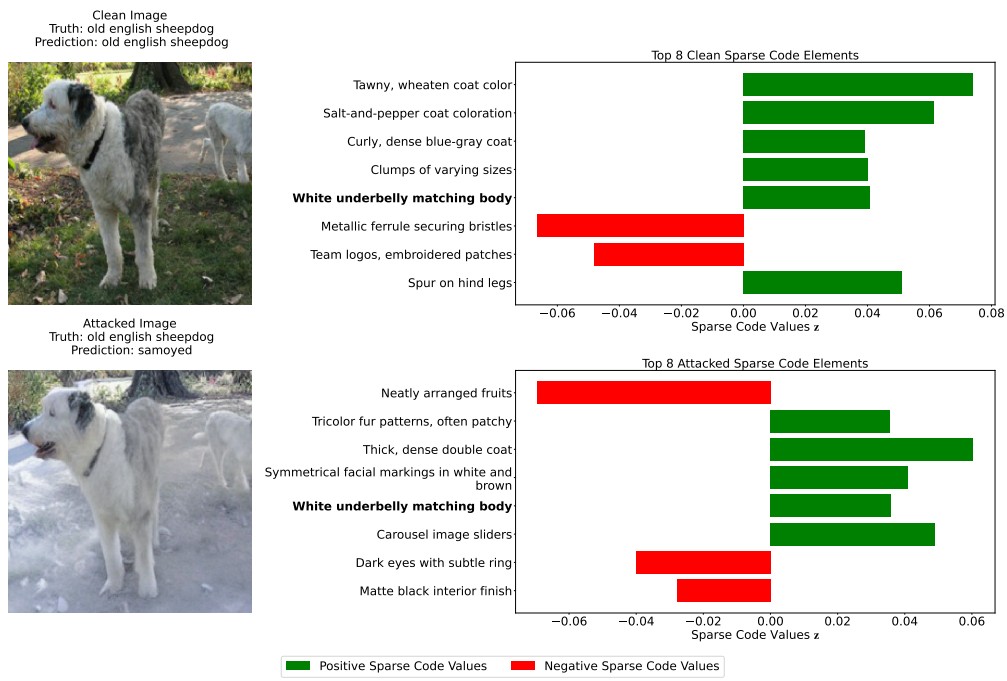

Figure 8: **The top concepts are unstable when the prediction is incorrect**. We visualize the top 8 concepts, using the same procedure as Figure 7.

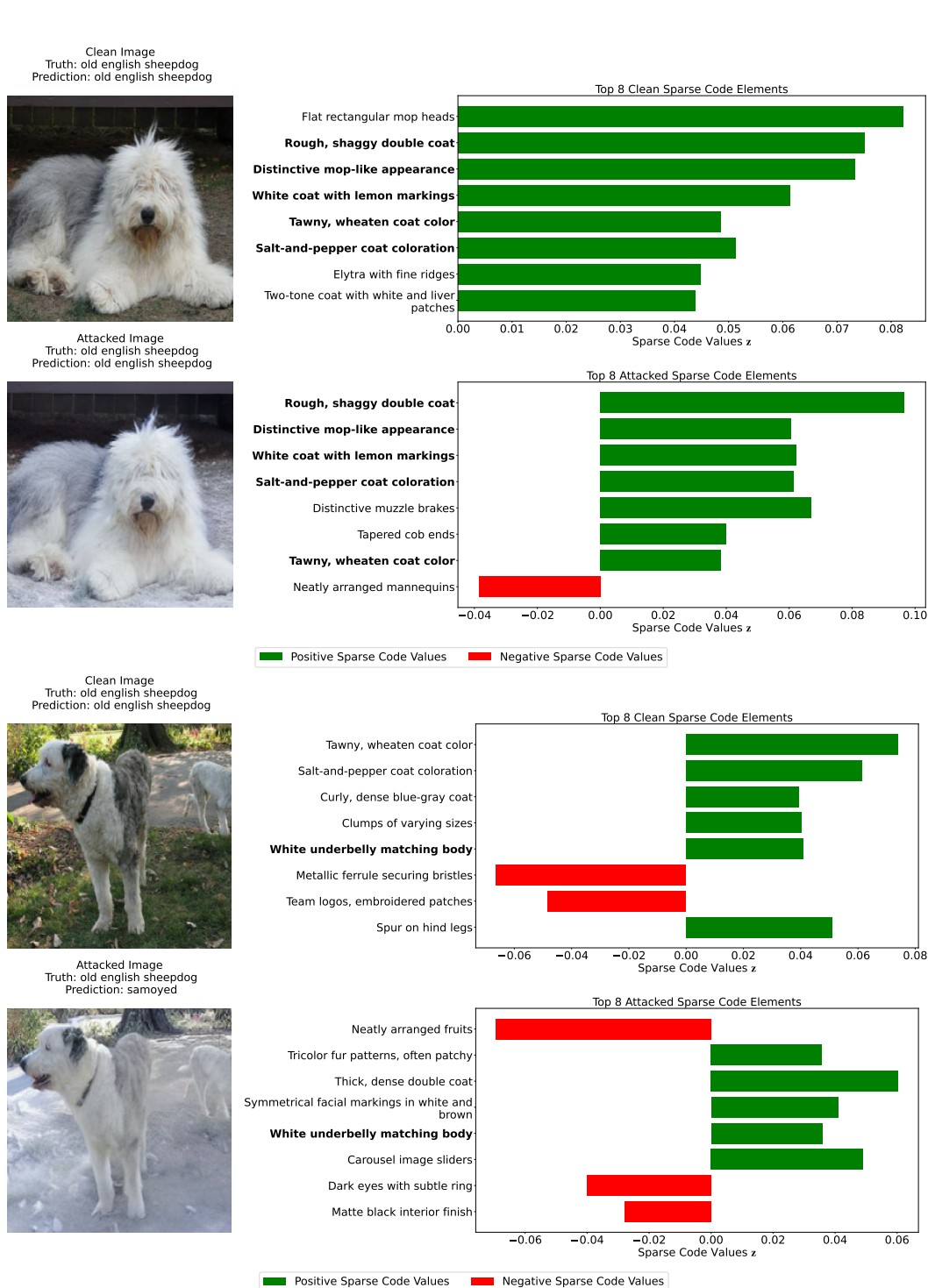

Figure 9: Correct and Incorrect classification examples from our dictionary (Section 3.2). We visualize the top 8 concepts, sorted based on the absolute value of the product of the sparse codes with the true label classifier weights. Concepts that are present in both the sparse code corresponding to the clean image and the sparse code corresponding to the adversarial image are in **bold**.

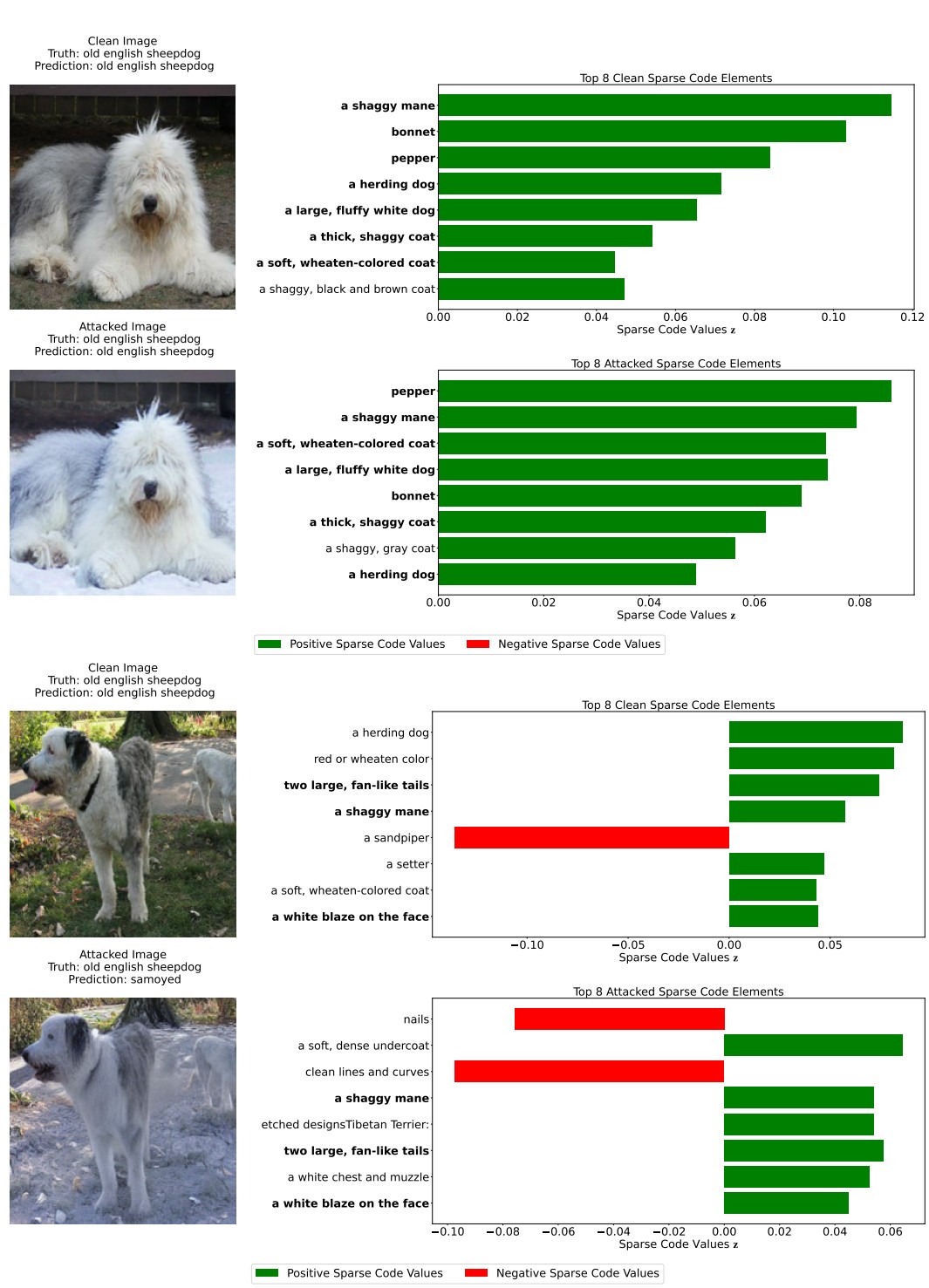

Figure 10: Correct and Incorrect classification examples from the Label-Free CBM (Oikarinen et al., 2023) dictionary. We visualize the top 8 concepts, sorted based on the absolute value of the product of the sparse codes with the true label classifier weights. Concepts that are present in both the sparse code corresponding to the clean image and the sparse code corresponding to the adversarial image are in **bold**.

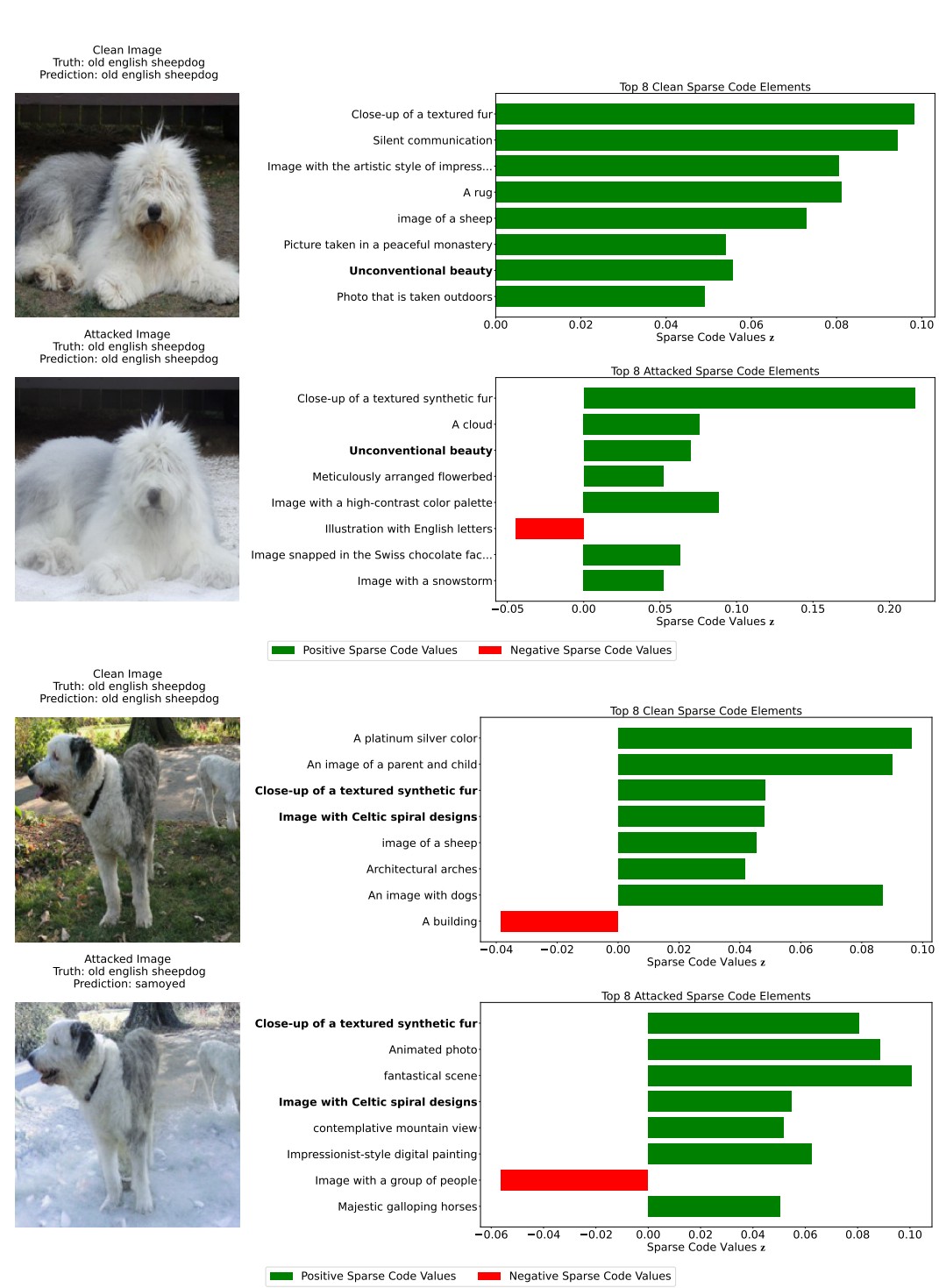

Figure 11: Correct and Incorrect classification examples from Gandelsman et al. (2024)'s dictionary. We visualize the top 8 concepts, sorted based on the absolute value of the product of the sparse codes with the true label classifier weights. Concepts that are present in both the sparse code corresponding to the clean image and the sparse code corresponding to the adversarial image are in **bold**.

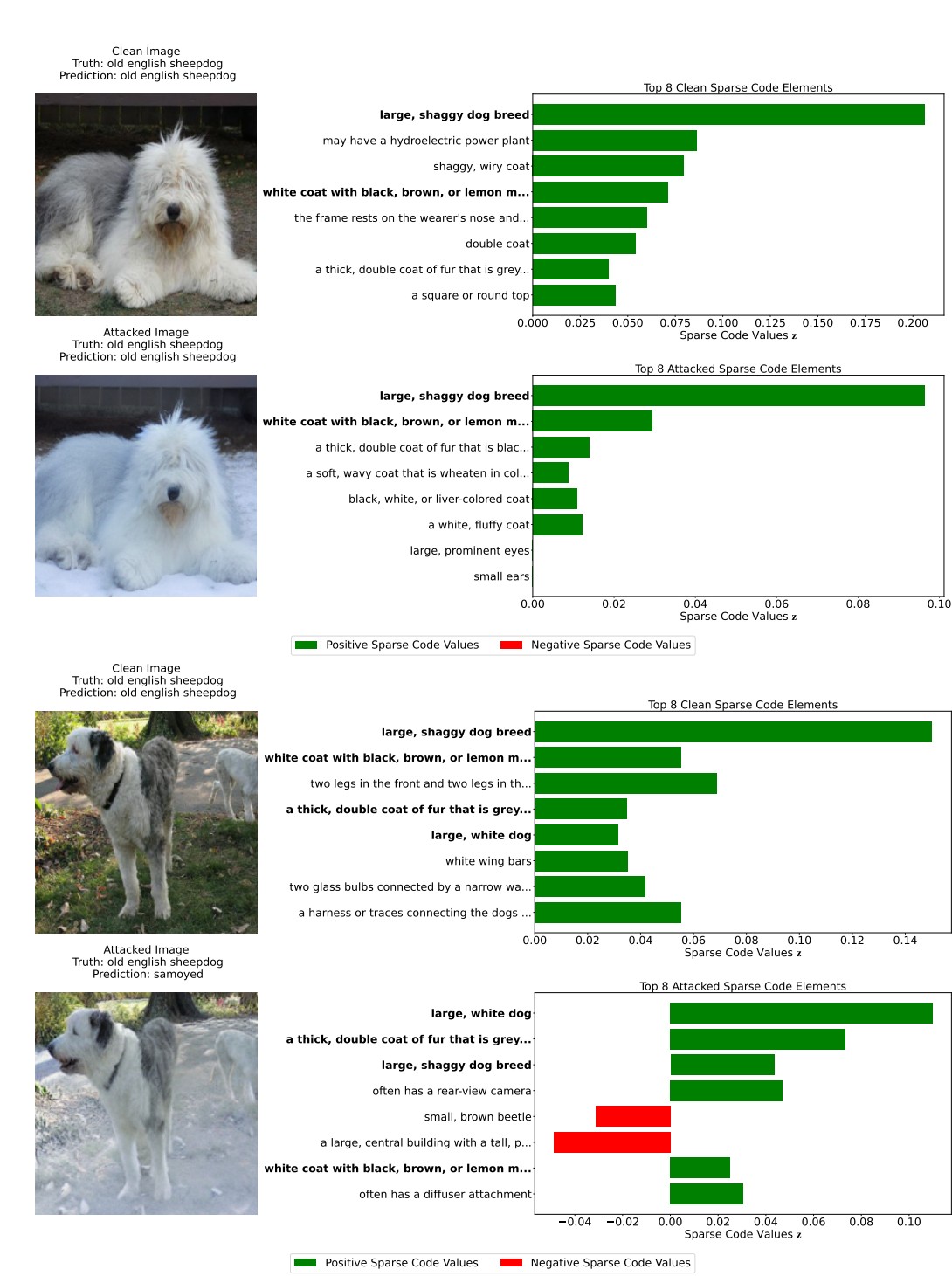

Figure 12: Correct and Incorrect classification examples from Chiquier et al. (2024). We visualize the top 8 concepts, sorted based on the absolute value of the product of the sparse codes with the true label classifier weights. Concepts that are present in both the sparse code corresponding to the clean image and the sparse code corresponding to the adversarial image are in **bold**.

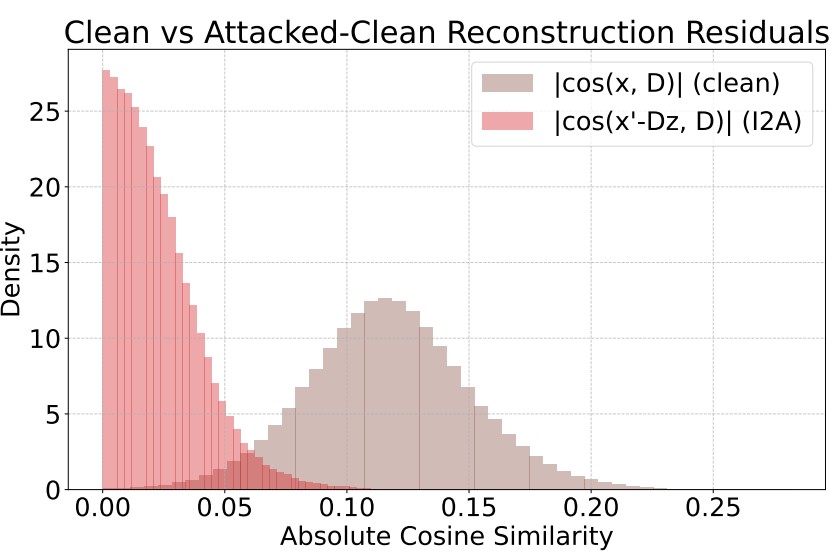

Figure 13: Analysis of the difference between attacked and clean reconstructions on ImageNet. This plot compares the residual $x' - \mathbf{D}z$ (attacked embedding minus the clean reconstruction) to demonstrate that the attack modifies directions not captured by the dictionary atoms, supporting the claim that the dictionary is expressive for relevant concepts and attacked directions are largely orthogonal to the span of $\mathbf{D}$.

