# OpenReview forum: "SSD: A Sparse Semantic Defense Against Semantic Adversarial Attacks to Image Classifiers"
_ICLR.cc/2026/Conference — Submitted to ICLR 2026_

### Official Review · Reviewer_HCHB · 2025-10-21

**Soundness:** 3
**Presentation:** 3
**Contribution:** 2
**Rating:** 6
**Confidence:** 3

**Summary:**

The paper defends in semantic space rather than in pixels: it builds a task-relevant concept dictionary with WordNet+GPT, embeds images/text with CLIP, then uses LASSO to get a sparse “concept code”; a linear head predicts from that code. The residual is meant to soak up irrelevant semantics/attacks. Results on ImageWoof/ImageNet against Instruct2Attack/ColorFool look solid for a training-free defense, and the method is naturally interpretable.

**Strengths:**

1. SSD is among the first works to explicitly design a semantic-space defense grounded in language–vision alignment.
2. Proposition 1 provides a clear rationale under RIP conditions for why sparse coding can separate relevant vs. irrelevant semantics, which is conceptually elegant.
3. Compared to adversarial training or diffusion-based defenses, SSD avoids expensive semantic adversarial generation and adds minimal inference overhead.

**Weaknesses:**

1. Current prompts (“make it in snow/at night”) cover appearance; consider adding other semantic editors to back Fig. 3’s orthogonality story.
2. The paper reports robust accuracy but provides no certified robustness (e.g., smoothing bounds). This matters because the defense is deterministic at test time and thus a natural target for adaptive optimization.
3. The paper references an exact prompt (Fig. 5) and a filtering step (Appx D.4), but the main paper doesn’t quantify final |C|, coverage, or diversity metrics; small changes in prompt/filtering could materially change robustness/interpretability.

**Questions:**

1. How large is the dictionary (|C|) on ImageNet, typical nonzeros per image, and per-image solve time for Eq. (3)?
2. How do you pick λ and the nonzero cap (100/200)?

---

> ### Author Response · Authors · 2025-11-24
>
> Dear Reviewer HCHB,
>
> Thank you for appreciating SSD’s novelty, theoretical rationale, and computational benefits compared to existing defenses. We appreciate your additional questions, which we address below. All updated results are in **Appendix E** of the revised PDF.
>
> 1. **(W1) Semantic Attack Diversity**
> - Here you go: we use the remaining attacks given by Instruct2Attack to compare the proposed SSD with baselines on ImageWoof in **Table 14 and Table 15**\. The proposed SSD is now robust across lighting (“make it at night”), weather (“make it in snow”), and image style (“make it a sketch painting”, “make it a vintage photo”) modifications.
>
> **Table 14: Comparison of standard and robust accuracy for different models on ImageWoof and ImageNet with adaptive attacks with the “make it a sketch painting” prompt.**
>
> | Dataset | Model | Standard Accuracy ↑ | Robust Accuracy ↑ |
> | :--- | :--- | :--- | :--- |
> | ImageWoof | CLIP (Radford et al., 2021) | **92.5** | 32.7 |
> | ImageWoof | FARE2-CLIP (Schlarmann et al., 2024) | 81.4 | *38.8* |
> | ImageWoof | CLIP + SSD (Ours) | *85.8* | **51.2** |
> | ImageNet | CLIP (Radford et al., 2021) | **82.7** | 4.44 |
> | ImageNet | FARE2-CLIP (Schlarmann et al., 2024) | 80.9 | *4.47* |
> | ImageNet | CLIP + SSD (Ours) | *81.7* | **9.40** |
>
>
> **Table 15: Comparison of standard and robust accuracy for different models on ImageWoof with adaptive attacks with the “make it a vintage photo” prompt.**
>
> | Dataset | Model | Standard Accuracy ↑ | Robust Accuracy ↑ |
> | :--- | :--- | :--- | :--- |
> | ImageWoof | CLIP (Radford et al., 2021) | **92.5** | 27.7 |
> | ImageWoof | FARE2-CLIP (Schlarmann et al., 2024) | 81.4 | *44.4* |
> | ImageWoof | CLIP + SSD (Ours) | *85.8* | **53.7** |
>
> - We note that curating a dataset beyond Instruct2Attack with more diverse attacks that are semantically irrelevant for the classification is a fascinating future direction, which is beyond the scope of the paper.
> 2. **(W2) Possible extension to certified robustness**
> - We agree that certified robustness is the ideal goal; however, several challenges exist. First of all, even formulating the space over which to certify robustness for semantic attacks is an open problem.
> - Furthermore, while efforts have been made by \[4\] to obtain robustness certifications for sparse coding algorithms, these certifications only hold for $\\ell\_2$ attacks.
> - Extending this to any other notion of semantic distance is an interesting direction for future work, but it is not the focus of our paper. Our contributions highlight an empirical defense that works well against semantic attacks.
> 3. **(W2) Lack of adaptive attack**
> - We would like to emphasize that the results for our method and CLIP, FARE-CLIP, and TeCoA-CLIP in **Table 1** and **Table 2** are *already* utilizing an **adaptive** attack. More precisely, the result for each model is evaluated against an attack adaptively optimized against that model.
> - Perhaps what caused confusion was the original writing in lines 369-372; we clarify that only the results in Table 4 (in the Appendix) are non-adaptive, allowing for a fair comparison with DiffPure. We have updated the text in the Experiments section to clarify this. Thank you for the catch.
> 4. **(W3, Q1, Q2) Hyperparameter Details**
> - **Dictionary metrics:** Originally, we provided the dictionary size |C| before and after each filtering step in Table 12\. We are happy to provide updated metrics relating to coverage and diversity in **Table 12**. For simplicity, we chose the default filtering hyperparameters from Label-Free CBM \[1\] and found that the filtered concepts are reasonable upon manual inspection.
> - **Nonzeros per image:**
>     - The SPAMS library \[2\] implementation provides a maximum number of iterations, so for simplicity, we pick a small value for $\\lambda$ and vary the maximum number of iterations $k$.
>     - To determine the value of $k$, we conducted an ablation study, which is presented in **Figure 6**.
> - **Solve time per image:** SSD takes **0.26 seconds per image**. In comparison, DiffPure \[3\] takes **11.2 seconds per image**.
>
> We appreciate the constructive feedback and are happy to engage in further discussions.
>
> Best regards,
> Authors of Submission \#4860
>
> \[1\] Oikarinen, Tuomas P., et al. "Label-free Concept Bottleneck Models." ICLR. 2023\.
> \[2\] J. Mairal, F. Bach, J. Ponce, et al. Sparse modeling for image and vision processing. Foundations and Trends® in Computer Graphics and Vision, 8(2-3):85–283, 2014\.
> \[3\] Nie, Weili, et al. "Diffusion Models for Adversarial Purification." International Conference on Machine Learning. PMLR, 2022\.

---

### Official Review · Reviewer_ajef · 2025-10-27

**Soundness:** 2
**Presentation:** 2
**Contribution:** 1
**Rating:** 0
**Confidence:** 5

**Summary:**

The paper proposes Sparse Semantic Defense (SSD): represent each image as a sparse combination of task-relevant, text-encoded concepts and classify using those sparse codes. Off-dictionary perturbations are absorbed into a residual, keeping key concept coefficients stable—improving robustness to semantic edits without adversarial training and yielding concept-level interpretability. Limitations include reliance on the concept dictionary’s quality/coverage and limited testing against fully adaptive semantic attacks.

**Strengths:**

This method doesn’t rely on ℓp bounds and holds up across varied semantic edits (weather, lighting, color)，aligning better with real-world shifts.

**Weaknesses:**

Although the method proposed in this paper involves a series of complex components and processes including GPT and CLIP，however, I believe it is meaningless. For classifier defenses, there are already many simple and effective approaches such as image pre-processing and adversarial training methods, all of which consume far fewer resources than the method in this paper. Even if one insists on leveraging large language models, there exist numerous simple yet effective solutions.

For example, I directly input the adversarial image from Figure 1 into GPT-5. GPT-5 not only produced the correct classification but also provided a clear and interpretable explanation—demonstrating better interpretability than the method proposed in this paper. Below, I quote GPT’s answer:

“Q: Tell me the name of this kind of dog

GPT-5 Answer: Looks like a Shih Tzu—a small, flat-faced toy breed with a long, fluffy double coat. (Sometimes people mix them with Lhasa Apsos, but the shorter muzzle and rounder head point to Shih Tzu.)”

Thus, since GPT alone can already achieve the defense, the series of so-called post-processing steps proposed by the authors on top of GPT become entirely unnecessary. The proposed method is not only much less efficient than traditional adversarial defense techniques, but also less effective than simply using GPT.

In summary, compared to existing approaches, this paper’s method offers no advantages and instead increases computational cost. So I believe this work is meaningless.

**Questions:**

See the weakness part of my review.

---

> ### Author Response · Authors · 2025-11-24
>
> Dear Reviewer ajef,
>
> Thank you for appreciating how SSD performs across different edits. We respectfully disagree with the weaknesses, as elaborated below.
>
> 1. *“there are already many simple and effective approaches such as image pre-processing and adversarial training methods”*
> - Choosing tractable adversarial training algorithms using $\\ell\_2$ adversaries (such as FARE-CLIP \[1\] and TeCoA-CLIP \[2\]) is less effective at robustness, as evidenced in **Table 1** and **Table 2** in the main paper. Our results show that SSD outperforms these $\\ell\_2$ adversarial training baselines (e.g., SSD improves robust accuracy by up to 43% relative to $\\ell\_2$ training on ImageNet against Instruct2Attack).
> - Regarding image-preprocessing techniques, we already show in Table 6 of the main paper that SSD is *competitive* with a popular image pre-processing method, DiffPure \[3\], while being significantly **more efficient** (0.26 seconds/image vs. 11.2 seconds/image).
> 2. *“all of which consume far fewer resources than the method in this paper”*
> - As mentioned in the introduction (lines 81-83), adversarial training against semantic attacks is extremely **expensive** in practice due to the computational cost of generating rich semantic attacks. We will also explain this further here.
>   - Generating one adversarial image for Instruct2Attack on one prompt against a CLIP classifier takes 73.8s, a 246$\\times$ increase in time versus generating an adversarial image with an $\\ell\_2$ PGD attack (0.3s). Adversarial training may require multiple iterations, thereby multiplying the computational cost by the number of iterations.
>   - For each new possible modification/prompt, one would need to redo adversarial training for this modification, increasing the cost of adversarial training combinatorially.
> 3. *“For example, I directly input the adversarial image from Figure 1 into GPT-5.”*
> - This is not an apple-to-apple comparison, for two following reasons.
>   - The image in Figure 1 is adversarially generated by a prior work Instruct2Attack \[4\] to attack the CLIP encoder. As such, the image is not expected to fool other classifiers including GPT-5.
>   - The paper considers a white-box attack, where the attacker has full-access to all the defense methods (i.e., the ability to evaluate the classifier and its gradient with respect to the input image). In sharp contrast, it is completely different to attack (and defend) GPT-5, since one does not have its gradients due to its proprietary and closed-source nature.
> 4. *“the method proposed in this paper involves a series of complex components and processes including GPT and CLIP”*
> - We do not agree that SSD is a series of complex steps, as the conceptual framework is simply decomposing an image as a sparse linear combination of concepts in a semantic space (the CLIP latent space is an instance).
> - We quote the following comments from other reviewers, which praise the simplicity of SSD:
>   - "A clean, interpretable take on semantic defenses, and the overall idea feels fresh." (**Vouy**)
>   - "Simple and intuitive, combining LLM-generated concepts and sparse coding for enhanced interpretability." (**Vouy**)
>   - "Conceptually elegant." (**HCHB**)
>
>
>
> Thank you for your efforts. We are happy to engage in further discussions.
>
> Best regards,
> Authors of Submission \#4860
>
> \[1\] Schlarmann, Christian, et al. "Robust CLIP: Unsupervised Adversarial Fine-Tuning of Vision Embeddings for Robust Large Vision-Language Models." International Conference on Machine Learning. PMLR, 2024\.
> \[2\] Mao, Chengzhi, et al. "Understanding Zero-shot Adversarial Robustness for Large-Scale Models." International Conference on Learning Representations. PMLR, 2023\.
> \[3\] Nie, Weili, et al. "Diffusion Models for Adversarial Purification." International Conference on Machine Learning. PMLR, 2022\.
> \[4\] Liu, Jiang, et al. "Instruct2attack: Language-guided semantic adversarial attacks." arXiv preprint arXiv:2311.15551 (2023).

---

> > ### Comment · Reviewer_ajef · 2025-11-26
> >
> > I have read your rebuttal carefully. While I acknowledge the experimental efforts demonstrating SSD's improvement over specific baselines like DiffPure or adversarial training on CLIP, my fundamental concern regarding the **practical significance and necessity** of this work remains unresolved.
> >
> > **1. The "Paradigm Shift" in Robustness:**
> > You argued that comparing against GPT-5 is not "apple-to-apple" because the attack was optimized for CLIP. However, this actually reinforces my point. Semantic adversarial attacks (e.g., changing weather to "snow") are defined as generating naturally looking, content-preserving perturbations.
> > * The fact that a Foundation Model (like GPT-4V/5) correctly classifies the image without any specific defense mechanism proves that **robustness to semantic shifts is an emergent property of better, larger models.**
> > * Therefore, proposing a complex pipeline (GPT-generated dictionary $\rightarrow$ CLIP encoder $\rightarrow$ LASSO sparse coding optimization $\rightarrow$ classifier) to "patch" the vulnerabilities of a weaker encoder (CLIP) feels like a step backward. The field is moving towards inherently robust Large Multimodal Models (LMMs), making ad-hoc defense modules for CLIP less relevant.
> >
> > **2. Inference Latency and Efficiency:**
> > Regarding efficiency, even if we accept the 0.26s/image figure mentioned in your rebuttal (which differs from the 0.81s reported in Table 4 of your paper), this still represents a **25x-80x slowdown** compared to the standard CLIP inference (0.01s).
> > * In real-world deployment, increasing latency by such a magnitude is often unacceptable.
> > * While SSD is faster than the extremely slow DiffPure, it is significantly less efficient than simply using a more capable model or standard data augmentation strategies that do not incur inference-time optimization costs.
> >
> > **3. Complexity vs. Utility:**
> > My critique on complexity is not just about the mathematical formulation, but the **engineering dependency**. The method relies on the alignment between an LLM (for dictionary generation) and a VLM (CLIP), plus a sparse solver. This introduces multiple points of failure (e.g., dictionary quality, alignment issues mentioned in your limitations). Compared to the "simple and effective" baseline of just using a stronger model (or even standard adversarial training which allows fast inference), the trade-off offered by SSD does not seem favorable.

---

> > > ### Author Response · Authors · 2025-11-30
> > >
> > > Thank you for your reply. We still **disagree** that your comparison is appropriate.
> > >
> > > 1. **"Paradigm Shift in Robustness” is just ignoring weaknesses**:
> > > - The purpose of generating an attack optimized for a model is to discover hidden vulnerabilities in that model. Our attacks are generated on CLIP, so CLIP will be affected the most. If we have the capacity to generate attacks on GPT-5 (white-box access, compute), we can also discover attacks that degrade GPT-5 performance.
> > > - To illustrate that GPTs are not perfect, \[5\] have found that changing picture styles (without any optimization, such as Instruct2Attack) already **degrades** the performance of GPT-4 from 81.5 to 62.6 (see Table 1 of \[5\]).
> > > - How much worse would the result be if the attacks are optimized to find actual weaknesses?
> > > 2. **We disagree that** *“to "patch" the vulnerabilities of a weaker encoder (CLIP) feels like a step backward”*
> > > - Instead of hoping that larger models and more training will automatically make the model robust, we return to the foundation and ask what is necessary to robustify a model.
> > > - We are not opposed to using large models, but we believe that a smaller model provides a quick feedback mechanism to help us discover how to make models robust. If we cannot robustify CLIP, how can we robustify the larger GPT-5 model (whose white-box access is not even accessible to everyone)?
> > > 3. *“**Inference Latency and Efficiency”*** **is more complicated:**
> > > - **First**, there is no official number available for the inference time per image for GPT-5, so arguing that using a large model will be more efficient is **unfounded**.
> > >     - The average inference time/image by GPT-5 through the API is 1.265 $\\pm$ 0.388 seconds (on 100 images) (compared with 0.81 of using CLIP + our method SSD)
> > > - We apologize for the typo; the correct number is 0.81, not 0.26. Thank you for catching this.
> > > - **Second**, please note that the 0.8 seconds per image (0.81 \- 0.01) is merely an overhead for the sparse coding step in the embedding space and does not increase as the image encoder becomes larger. If the inference time of a larger image encoder increases, the overhead for sparse coding will be less noticeable.
> > > 4. “standard adversarial training which allows fast inference” **is hiding the cost away at training and might not be effective**
> > > - We would like to repeat points 1 and 2 made in our first reply, that adversarial training on $\\ell\_2$ attack is **not effective,** and semantic adversarial attacks are simply **combinatorially expensive**.
> > > 5. *“relies on the alignment between an LLM (for dictionary generation) and a VLM (CLIP)”* **is wrong**
> > > - Our method does not rely on the alignment between an LLM and a VLM.
> > > - *As shown in* **Figure 1** of the paper, the LLM (GPT-4 in the paper) enables us to generate natural language concepts at scale (we can also derive natural language concepts from human supervision if needed). We then take these natural language concepts and use the text encoder in CLIP to generate the dictionary of concept embeddings.
> > > 6. **Complexity vs. Utility:**
> > > - First, we want to iterate that the robustness of a larger model is not guaranteed, as mentioned in point 1 of this reply.
> > > - The “simple and effective” baseline of using a stronger model comes with its own **financial complexity**.
> > >   - For each new image, we would need to use the GPT-5 API, and the cost does not scale well if we have a large number of images.
> > >   - On the other hand, we use GPT only once to generate natural language concepts before utilizing the CLIP text encoders to create the dictionary.
> > >   - As an analogy, if $N$ is the number of images, the cost of using the GPT-5 API is $O(N)$, while the cost of our approach is $O(1)$.
> > > - Also, in certain applications (e.g., medical imaging), it is illegal to “simply use a better model” due to HIPAA regulations. If that is true, knowing how to robustify your internal model is a must. Please note that we never show GPT any of the images when generating the dictionary of concept embeddings.
> > >
> > > \[5\] Cai, Rizhao, et al. "Benchlmm: Benchmarking cross-style visual capability of large multimodal models." European Conference on Computer Vision. Cham: Springer Nature Switzerland, 2024\.

---

### Official Review · Reviewer_Vouy · 2025-10-30

**Soundness:** 2
**Presentation:** 3
**Contribution:** 3
**Rating:** 6
**Confidence:** 3

**Summary:**

This paper proposes Sparse Semantic Defense (SSD), a defense method against semantic attacks. The main idea is to use a language model to generate class-related visual concepts, encode them with a CLIP text encoder to form a semantic dictionary, and then decompose image features into a sparse combination of these concepts via LASSO. The approach enhances robustness while maintaining interpretability. The paper also provides a theoretical explanation and demonstrates improved robustness on ImageWoof and ImageNet. It’s a clean, interpretable take on semantic defenses, and the overall idea feels fresh.

**Strengths:**

1. The method is simple and intuitive, combining LLM-generated concepts and sparse coding for enhanced interpretability.

2. SSD shows consistent robustness gains under several semantic attacks and is more efficient at inference compared to diffusion-based defenses.

3. Theoretical intuition and empirical evidence together help clarify why sparse semantic representation improves robustness.

**Weaknesses:**

1. Theoretical assumptions (orthogonality and RIP) are strong and not validated on larger or more realistic datasets.

2. Experimental coverage is limited, i.e. semantic attack diversity and adaptive comparisons are lacking.

3. The concept dictionary still contains noisy or irrelevant entries, weakening interpretability claims.

**Questions:**

1. Could the authors validate the "orthogonality between attack residuals and dictionary space" assumption on a larger dataset, even with simple statistics?

2. Would it be possible to design a lightweight approximation of adaptive attacks to more fairly test SSD against stronger defenses?

3. Could the authors show robustness results under additional prompts (e.g., different scene or style edits) to demonstrate generalization?

---

> ### Author Response · Authors · 2025-11-24
> **Official Comment (1/2)**
>
> Dear Reviewer Vouy,
>
> Thank you for acknowledging the simplicity and efficiency of SSD, along with theoretical intuition and empirical evidence on why sparse representation improves robustness. We appreciate your questions about extra validation, which we will address below. All updated results are in **Appendix E** of the revised PDF.
>
> 1. **(W1, Q1) Validity of theoretical assumptions**
>    Thank you for the constructive comment\! Below we provide insights and revision of the paper both **empirically** and **theoretically**.
> **Empirical Validation**:
> - We kindly refer to Figure 3 of the original paper for a verification on 1000 images from the 10 classes of ImageWoof.
> - That said, we are happy to conduct additional experiments to validate orthogonality on ImageNet and provide the results in **Figure 13**.
>   - As seen, orthogonality between attack residuals and the dictionary atoms holds approximately on a larger dataset. Thank you for the comment that helps strengthen the paper!
>   - Due to the constraints on compute and time for generating the attacks using Instruct2Attack, Figure 13 is produced on the 5000-image test subset used by RobustBench with 1000 classes.
> **Revision of theory**:
> - We see your point, and yes *exact* orthogonality is a strong assumption. As shown in **Figures 3** **and 13**, the attack residual is *approximately* orthogonal to the dictionary. Inspired by your comment and the empirical observation, we note that a theorem that depends on the incoherence (which quantifies the approximate orthogonality) is possible, as shown in Theorem 4.1 of \[1\]. We  have updated the paper to include this reference; thank you\!
> 2. **(W2) Adaptive comparisons**
> - We would like to clarify that the results for our method, CLIP, FARE-CLIP \[2\], and TeCoA-CLIP \[3\] in **Table 1** and **Table 2** are *already* utilizing an **adaptive** attack. More precisely, the result for each model is evaluated against an attack adaptively optimized against that model.
> - We hear you. Perhaps what caused confusion was the original writing in lines 369-372. We clarify that only the results in Table 4 of the Appendix are non-adaptive, which were included for a fair comparison with DiffPure. We have updated the text in the Experiments section to make it clear. Thank you for the catch\!
> 3. **(W2, Q3) Semantic attack diversity**
> - Here you go: we use the remaining attacks given by Instruct2Attack to compare the proposed SSD with baselines on ImageWoof in **Table 14 and Table 15**\. The proposed SSD is now robust across lighting (“make it at night”), weather (“make it in snow”), and image style (“make it a sketch painting”, “make it a vintage photo”) modifications.
>
> **Table 14: Comparison of standard and robust accuracy for different models on ImageWoof and ImageNet with adaptive attacks with the “make it a sketch painting” prompt.**
>
> | Dataset | Model | Standard Accuracy ↑ | Robust Accuracy ↑ |
> | :--- | :--- | :--- | :--- |
> | ImageWoof | CLIP (Radford et al., 2021) | **92.5** | 32.7 |
> | ImageWoof | FARE2-CLIP (Schlarmann et al., 2024) | 81.4 | *38.8* |
> | ImageWoof | CLIP + SSD (Ours) | *85.8* | **51.2** |
> | ImageNet | CLIP (Radford et al., 2021) | **82.7** | 4.44 |
> | ImageNet | FARE2-CLIP (Schlarmann et al., 2024) | 80.9 | *4.47* |
> | ImageNet | CLIP + SSD (Ours) | *81.7* | **9.40** |
>
> **Table 15: Comparison of standard and robust accuracy for different models on ImageWoof with adaptive attacks with the “make it a vintage photo” prompt.**
>
> | Dataset | Model | Standard Accuracy ↑ | Robust Accuracy ↑ |
> | :--- | :--- | :--- | :--- |
> | ImageWoof | CLIP (Radford et al., 2021) | **92.5** | 27.7 |
> | ImageWoof | FARE2-CLIP (Schlarmann et al., 2024) | 81.4 | *44.4* |
> | ImageWoof | CLIP + SSD (Ours) | *85.8* | **53.7** |
>
> - We note that curating a dataset beyond Instruct2Attack with more diverse attacks that are semantically irrelevant for the classification is a fascinating future direction, which is beyond the scope of the paper.

---

> > ### Author Response · Authors · 2025-11-24
> > **Official Comment (2/2)**
> >
> > 4. **(W3) Some irrelevant entries affecting interpretability**
> > - Akin to standard sparse coding, our dictionary is assumed to be overcomplete and contains a large collection of concepts. For a given adversarial image, our interpretable classification is provided by the sparse selection of concepts from our algorithm.
> > - Our algorithm naturally gives interpretability as the cosine similarity between an image and concepts irrelevant to that image would be small, yielding small absolute coefficients in the resulting sparse code. From this perspective, a large concept dictionary does not harm interpretability but rather allows stronger generalization to a rich variety of input images.
> >
> > We appreciate the constructive feedback and are happy to engage in further discussions.
> >
> > Best regards,
> > Authors of Submission \#4860
> >
> > \[1\] Cai, T. Tony, Guangwu Xu, and Jun Zhang. "On Recovery of Sparse Signals Via $\\ell_{1}$ Minimization." IEEE Transactions on Information Theory 55.7 (2009): 3388-3397.
> > \[2\] Schlarmann, Christian, et al. "Robust CLIP: Unsupervised Adversarial Fine-Tuning of Vision Embeddings for Robust Large Vision-Language Models." International Conference on Machine Learning. PMLR, 2024\.
> > \[3\] Mao, Chengzhi, et al. "Understanding Zero-shot Adversarial Robustness for Large-Scale Models." International Conference on Learning Representations. PMLR, 2023\.

---

### Author Response · Authors · 2025-11-24
**Global Response**

Thank you the AC and the reviewers for their efforts. Below we summarize the reviews and what we have done to address the comments.

We appreciate that the reviewers consistently praised the paper's novelty, interpretability, and efficiency.

1. **Novelty & Simplicity:**
- "A clean, interpretable take on semantic defenses, and the overall idea feels fresh." (**Vouy**)
- "Simple and intuitive, combining LLM-generated concepts and sparse coding for enhanced interpretability." (**Vouy**)
- "Conceptually elegant." (**HCHB**)
- "Among the first works to explicitly design a semantic-space defense grounded in language–vision alignment." (**HCHB**)
2. **Robustness:**
- "Shows consistent robustness gains under several semantic attacks." (**Vouy**)
- "Holds up across varied semantic edits (weather, lighting, color), aligning better with real-world shifts." (**ajef**)
3. **Efficiency:**
- "Avoids expensive semantic adversarial generation and adds minimal inference overhead." (**HCHB**)
- "More efficient at inference compared to diffusion-based defenses." (**Vouy**)

We have addressed the primary concerns regarding validation, generalization, and comparisons through the following updates:

1. **Strengthened Theoretical & Empirical Validation**
- **Orthogonality Check:** We have added **Figure 13** (Appendix E) to verify the orthogonality between the attack residuals and the dictionary space on ImageNet (**Vouy**).
- **Theoretical Grounds:** We have revised the paper to note that weakening the exact orthogonality condition in Proposition 1 to approximate orthogonality (through incoherence) can be done through an existing work in the literature (**Vouy**).
2. **Expanded Test**
- **New Prompts**: We have added robustness results for additional Instruct2Attack prompts on ImageWoof & ImageNet in **Tables 14 & 15** (**Vouy, HCHB**).
3.  **Clarified Comparisons & Baselines**
- **Adaptive Attacks:** We confirm that the results in **Tables 1 & 2** already reflect adaptive attacks optimized against the specific model (**Vouy**, **HCHB**).
- **Dictionary Metrics:** We have included quantitative metrics on coverage and diversity in **Table 12** to address concerns about filtering and quality (**HCHB**).
- **Efficiency Defense:** We have highlighted the speed advantage (0.26s vs. 11.2s per image) and performance competitiveness against diffusion-based methods like DiffPure (**ajef**, **HCHB**).
- **Irrelevant GPT-5 Comparison:** We have clarified the scope (defending standard CLIP encoders vs. proprietary models) to state that it is not relevant to compare our method to GPT-5 (**ajef**).
4. **Technical Clarifications**
- **Sparsity Hyperparameters:** We have explained the selection of $\\lambda$ and $k$ via the ablation study in **Figure 6** (**HCHB**).
- **Irrelevant Entries:** We have clarified that low cosine similarity prevents irrelevant dictionary entries from affecting the sparse coding outcome (**Vouy**).

### Summary

In the individual rebuttals, we have addressed the reviewers’ valuable suggestions on expanded theoretical & empirical validation and clarification on the experiment details. All insights and questions are important for continuous improvements in our work. Based on this feedback, we have made and will make multiple revisions of the manuscript as promised.
In summary, we are grateful to have received thoughtful and insightful questions from the reviewers. Please don't hesitate to contact us for further discussion. Thank you all.

---

### Meta-Review · Area_Chair_M52A · 2025-12-11

**Summary:**

The idea of this paper is promising, and the problem it addresses (robustness under semantic adversarial attacks) is important. Unfortunately, the current work has several shortcomings that prevent it from reaching the acceptance threshold. Specifically, **first**, the experimental evaluation is limited. The authors only consider CLIP models with a ViT-L/14 backbone. For a study on robustness, this is not sufficient. There are many variants of CLIP models, and I suggest that the authors expand their experiments to include other visual backbones and other training paradigms, such as HyCoCLIP. **Second**, the authors’ response to Reviewer ajef is unconvincing. The reviewer’s fundamental concern is whether this study, limited to CLIP, remains meaningful in the era of large vision-language models. Responding that GPT-5 is closed-source provides little new insight. I recommend that the authors conduct further experiments on open models such as Qwen-2.5-7B. If the method can be successfully extended, the chances of acceptance in a future submission would greatly increase. **Third**, the concerns raised by Reviewers Vouy and HCHB regarding semantic attack diversity are also valuable and should be addressed more thoroughly. The authors should not be discouraged; the process of addressing these issues will provide important experience for your future research.

**Reviewer Concerns:**

**Reviewer Vouy** Reviewer Vouy’s main points were addressed by the authors through additional empirical validation of the orthogonality assumption on ImageNet, clarification that the primary experiments already use model-adaptive attacks, and expanded evaluation of semantic attack diversity with more Instruct2Attack transformations. Remaining concerns are minor, mainly the limited scope of the new validations, but overall most of Vouy’s comments were reasonably addressed.

**Reviewer ajef** The authors responded to Reviewer ajef’s points by demonstrating that SSD outperforms several adversarial-training baselines, clarifying efficiency relative to DiffPure, and explaining why comparisons to GPT-5 are not directly comparable under white-box threat models. Reviewer ajef’s fundamental concerns about the practical necessity of SSD given the robustness of modern multimodal models, the inference-time overhead, and the trade-off between complexity and utility remain unresolved.

**Reviewer HCHB** Reviewer HCHB’s questions were addressed through expanded semantic attack evaluations covering additional lighting, weather, and style edits, clarification that the main experiments use adaptive attacks, and more details on hyperparameters and the dictionary. Some concerns, such as the lack of certified robustness and the potential for even broader semantic attack coverage, remain outside the scope of the current paper.

**Reviewer Scores:**

**Reviewer Vouy**: Based on the content and tone of the review, I expect that Vouy would likely have maintained their original score had they been able to participate in the discussion. These comments are balanced, with moderate strengths and weaknesses noted, but none of the raised issues appear to be decisive or strongly argued. The concerns (e.g., theoretical assumptions, limited attack diversity, noisy dictionary entries) are valid yet relatively mild and do not suggest that the reviewer would update their assessment substantially in either direction. Overall, this reviewer’s evaluation seems stable, and no major points raised during the discussion would have materially shifted their score.

**Reviewer ajef**: The reviewer expressed strong and fundamental concerns about the practical significance of the proposed approach, repeatedly emphasizing that the method is unnecessarily complex, less efficient than existing defenses, and ultimately less meaningful than simply using stronger models (e.g., GPT-5). These concerns remained unchanged after reading the authors’ rebuttal. It is unlikely that ajef would have meaningfully revised the assessment during the discussion. I therefore expect that ajef would have maintained the original score, as their stance appears firmly held and not easily shifted by additional clarifications.

**Reviewer HCHB**: Reviewer HCHB provided a balanced and thoughtful review, outlining clear strengths of the proposed sparse semantic formulation as well as moderate concerns about evaluation breadth, dictionary characterization, and the lack of certified robustness. None of these points appear decisive or indicative of a major reassessment. Based on the tone and content of the review, I expect that Reviewer HCHB would have maintained the original score.

---

### Decision · Program_Chairs · 2026-01-26

Reject